# Visual Representations in Humans and Machines: A Comparative Analysis of Artificial and Biological Neural Responses to Naturalistic Dynamic Visual Stimuli

## Abstract

Visual representations in the human brain are shaped by the pressure to support planning and interactions with the environment. Do visual representations in deep network models converge with visual representations in humans? Here, we investigate this question for a new class of effective self-supervised models: Masked Autoencoders (MAEs). We compare image MAEs and video MAEs to neural responses in humans as well as convolutional neural networks. The results reveal that representations learned by MAEs diverge from neural representations in humans and convolutional neural networks. Fine-tuning MAEs with a supervised task improves their correspondence with neural responses but is not sufficient to bridge the gap that separates them from supervised convolutional networks. Finally, video MAEs show closer correspondence to neural representations than image MAEs, revealing an important role of temporal information. However, convolutional networks based on optic flow show a closer correspondence to neural responses in humans than even video MAEs, indicating that while masked autoencoding yields visual representations that are effective at multiple downstream tasks, it is not sufficient to learn representations that converge with human vision.

## 1 Introduction

Human vision is not an end in itself, but a means to an end. It has been shaped by evolutionary pressure to support our ability to interact with our surrounding environment (Lyon, 2007). This pressure has resulted in a visual system endowed with the ability to learn representations that can be used to perform a wide variety of tasks – from recognizing people to segmenting events, from estimating distances to detecting abnormalities in medical images. Machine vision aims to develop models with the ability to learn similarly flexible representations: Foundation Models of Vision (Awais et al., 2023). Testing the convergence between representations learned by machine vision models and representations in the human brain can offer a measure of the degree to which the models are approaching the human visual system.

A recent class of machine vision models – masked autoencoders (MAEs, He et al. (2022); Cao et al. (2022)) – have demonstrated a remarkable ability to support a variety of visual tasks. These models achieve high performance at object detection, object segmentation, and classification tasks (He et al., 2022). MAEs can be extended naturally to the processing of video inputs (Feichtenhofer et al., 2022; Tong et al., 2022; Wang et al., 2023a), yielding competitive performance on action classification (Feichtenhofer et al., 2022; Tong et al., 2022) and effective transfer of features to new datasets (Tong et al., 2022). More recently, MAEs have been used in conjunction with knowledge distillation techniques Hinton (2015); Gou et al. (2021) to learn representations that outperform vanilla MAEs on datasets such as the Something-Something V2 (Goyal et al., 2017), following an approach known as Masked Video Distillation (MVD, (Wang et al., 2023b)).

Given the effectiveness of MAEs at learning flexible visual representations, here we ask whether their representations converge with the representations in the human visual system. This work compared representations in image based and video based MAEs, as well as masked video distillation,

Table 1: Models of visual cortex

| Model | Input | Output | Training dataset | #Selected layers |
|---|---|---|---|---|
| Supervised static | image | object identity | Image-net | 11 |
| Supervised static | image | action identity | HAA-500 | 11 |
| Self-supervised dynamic | video | optic flows | HAA-500 | 11 |
| Self-supervised dynamic | video | optic flows | Kinetics-400 | 11 |
| Supervised dynamic | optic flow | action identity | HAA-500 | 11 |
| pre-trained Dino-v2 | image | image | Image-net | 12 |
| pre-trained Masked Autoencoder | (masked) image | (unmasked) image | Image-net | 12 |
| fine-tuned Masked Autoencoder | image | object identity | Image-net | 12 |
| pre-trained Masked Autoencoder | (masked) image | (unmasked) image | Kinetics-400 | 12 |
| pre-trained Masked Video Autoencoder | (masked) video | (unmasked) video | Kinetics-400 | 12 |
| fine-tuned Masked Video Autoencoder | video | action identity | Kinetics-400 | 12 |
| pre-trained Masked Video Distillation | (masked) video | MAE & VideoMAE high-level features | Kinetics-400 | 12 |

to neural responses in different parts of the human visual system. The convergence between MAEs and neural representations was compared to the the convergence between the latter and convolutional neural networks using image (He et al., 2016) and video (Zhu et al., 2019) inputs. All models were compared to fMRI responses in different visual streams and functional regions of interests, using as input a quasi-naturalistic video (the Forrest Gump movie, (Hanke et al., 2016)).

## 2 METHODS

### 2.1 VISION MODELS

To study representations of quasi-naturalistic visual stimuli, we used a variety of vision models, including feed-forward convolutional neural networks, as well as state-of-the-art foundation vision models. The models vary in architecture, learning objective, and training data (Table 1). Here, we propose an overview of the models. Training details for the HAA-trained CNNs are presented in supplementary materials. The trained versions of all other models are adopted from their official implementation repository. For model details, refer to the original papers.

**Supervised (sup) static net** is the spatial stream of the hidden two-stream convolutional neural network model Zhu et al. (2019). The sup static net has a resnet18 architecture and encodes static features of visual stimulus. Two versions of the model were included in the models' pool: one is trained on Image-Net Deng et al. (2009) and predicts object identity, and the other is trained on

HAA-500 action dataset Chung et al. (2021) and predicts action label. Both versions take a single frame as input.

**Self-supervised (s-sup) dynamic net** is the first part of the temporal stream (i.e., motion net) in the hidden two-stream convolutional neural network model Zhu et al. (2019). The self-supervised dynamic net takes 11 consecutive frames as input and infers the optic flow between each pair of consecutive frames. The network is trained to minimize an self-supervised learning objective obtained by combining three loss functions:1) a pixel-wise reconstruction error, 2) a smoothness loss addressing the ambiguity problem of optic flow estimation (also known as the aperture problem), and 3) a structural dissimilarity between the original and the reconstructed image patches (see Zhu et al. (2019) for details of loss functions). The models' pool contains two versions of the self-supervised dynamic net, trained on the HAA-500 Chung et al. (2021), and Kinetics-400 Kay et al. (2017) action datasets Chung et al. (2021).

**Supervised (sup) dynamic net** is the second part of the temporal stream in the hidden two-stream convolutional neural network model Zhu et al. (2019). The model has resnet18 architecture and takes optic flows from the self-supervised dynamic net as input. We used the HAA-500 dataset Chung et al. (2021) and trained the supervised dynamic net to predict action labels using optic flows.

**Dino-v2** is a self-supervised vision model that uses self-distillation to learn robust visual features by optimizing a contrastive learning objective between a student and teacher network, each having a transformer architecture **?**. We included a pre-trained version of Dino-v2 trained on Imagenet Deng et al. (2009).

**Masked Autoencoders (MAE)** learn representations of the images they receive as input that can be used to reconstruct original uncorrupted images from corrupted (masked) input through a series of transformer blocks He et al. (2022). The models' pool contains three versions of the MAE model: 1) a pre-trained version, where the model is trained to reconstruct pixel values of each frame (image), 2) a fine-tuned version, where the pre-trained model is further fine-tuned to predict object identities from images and 3) a pre-trained version, where the model is trained to reconstruct pixel values of randomly masked space-time patches in a video Feichtenhofer et al. (2022). The first two versions were trained on Image-net Deng et al. (2009), and the third on Kinetics-400 Kay et al. (2017).

**Video Masked Autoencoder (VMAE)** learns a spatiotemporal representation of videos, required to reconstruct original uncorrupted videos, from corrupted (tube masked) input through a series of transformer blocks Tong et al. (2022). We added two versions of the VMAE to our models' pool. The first is a pre-trained version, where the model is trained to reconstruct missing pixels of the input set of frames. The second version is the fine-tuned version obtained by fine-tuning the pre-trained version to predict action labels of input videos. Both models take a consecutive set of frames as input, and were trained on the Kinetics-400 action dataset Kay et al. (2017).

**Masked Video Distillation (MVD)** learns a higher-level spatial and spatiotemporal representation of the input video, required to reconstruct the representation of teacher MAE and VMAE while taking corrupted (tube-masked) videos as input Wang et al. (2023b). Unlike VMAE and MAE, the MVD model does not use pixel-level errors as learning signals. Rather, it uses learning signals based on high-level features of the input video using pre-trained MAE and VMAE models' features as masked prediction targets. Using the Kinetics-400 action dataset Kay et al. (2017), a pre-trained version was obtained and added to the models' pool.

### 2.2 COMPARISON BETWEEN MODELS AND NEURAL RESPONSES

Models were compared to neural responses using Representational Dissimilarity Matrices (RDMs, Kriegeskorte et al. (2008)). In this study, RDMs are matrices whose rows and columns correspond to timepoints in the movie, such that the element of the matrix at a given row and column is the dissimilarity between the representation of the video at the timepoints that correspond to that row and column. Neural RDMs and model RDMs were compared by computing their Pearson correlation. The movie was divided into eight runs of similar length. The dimension of the RDMs obtained for the eight segments were 451, 441, 438, 488, 462, 439, 542, 338.

The match between neural RDMs and RDMs for an entire model were calculated by first computing RDMs for each layer of the model and then computing a linear combination of the layer RDMs that best matches the neural RDM. In order to prevent circularity in the analysis, the weights attributed to each layer in the linear combination were calculated using 7 of the 8 experimental runs, and were applied to the model RDMs in the left-out run to compute a "predicted" RDM. We then evaluated

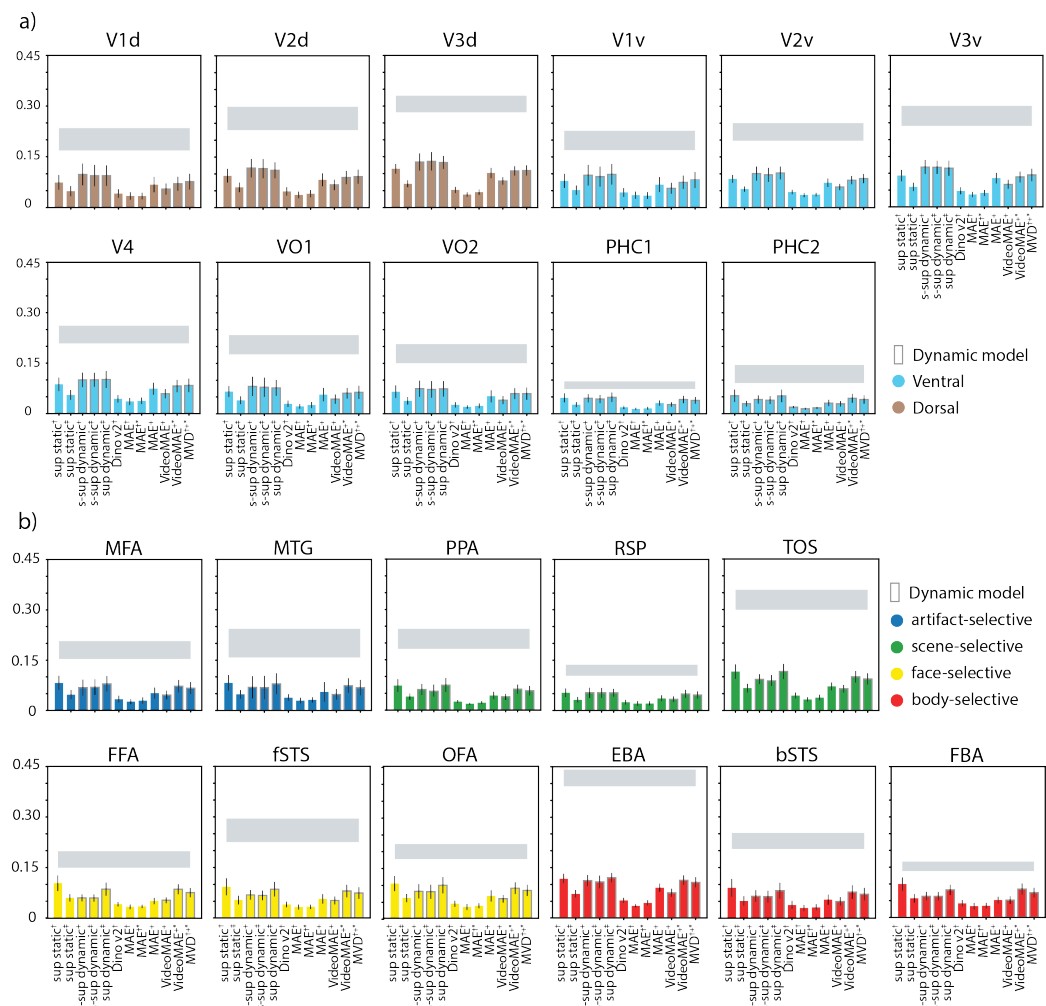

Figure 1: Pearson's correlation between actual and predicted brain regions' RDMs, averaged over participants for **a)** ventral and dorsal visual regions and **b)** visual category-selective regions. Predicted RDMs were obtained by training and testing a leave-one-out cross-validation linear regression model using a linear combination of each model's layers. Error bars show standard deviation over participants. Gray bands display noise ceiling. For each participant, the noise ceiling is calculated by averaging over all other participants prediction of the target participant's neural response (sup: supervised, s-sup: self-supervised, †: Image-net-trained, ‡: HAA-500-trained, +: Kinetics-400-trained, *: fine-tuned; MVD was trained on pre-trained MAE (Image-net) and VideoMAE (Kinetics-400))

the correlation between the "predicted" RDM and the neural RDM in the corresponding run (Figure 1).

To evaluate more directly the unique variance in a neural RDM that was explained by a model above and beyond each other model. To compute this, we regressed out a control model RDM from a neural RDM, and predicted the residual neural RDM with a target model, obtaining the unique variance explained by the target model. Matrices in Figure 2 show these difference values, with the target models as the columns and the control models as the rows.

## 3 RESULTS

The contribution of this work is to compare the representations in masked autoencoders (including video MAEs) to visual representations in the human brain. The human visual system learns a rich

set of visual representations, that enable us to perform a wide variety of tasks. Similarly, MAEs have been remarkably effective at a variety of tasks, ranging from object and action classification to segmentation He et al. (2022); Wang et al. (2023a). Previous work found that models with more accurate performance are also characterized by greater similarity with neural responses Yamins et al. (2014). If this phenomenon extends to MAEs, their effectiveness might make them more similar to the brain. Alternatively, comparing MAEs to the brain can reveal ways in which the models diverge from human vision.

The human visual system is organized into distinct regions with different response properties, including regions with selectivity for different object categories. This work takes into account the structure of the visual system, evaluating separately the correspondence between representations in different brain regions and the models. The first set of analyses (Figure 1) quantifies the correspondence between different models and visual as well as category-selective brain regions. For comparison purposes, we include multiple variants of MAEs as well as feedforward convolutional networks. The second set of analyses determine the extent to which each model explains unique variance in neural responses (Figure 2), that is not accounted for by other models. Finally, the third set of analyses study layer-to-layer variation in the models' representations. We identify dimensions that capture the differences between the representations in different layers and models, and search for interpretable properties that explain why different models vary in their correspondence with the brain.

## 3.1 SIMILARITY BETWEEN MODELS AND NEURAL RESPONSES

Representations of rich quasi-naturalistic video stimuli (the movie Forrest Gump) were extracted from masked autoencoders He et al. (2022); Cao et al. (2022), video masked autoencoders Feichtenhofer et al. (2022); Wang et al. (2023a), and masked video distillation Wang et al. (2023b). Representations of the same video stimuli were also extracted from a set of convolutional neural network models. The models varied along two key dimensions: 1) whether they encoded dynamic (hidden two-stream networks Zhu et al. (2019), video-masked autoencoders Wang et al. (2023a), and masked video distillation Wang et al. (2023b)) or static (standard convolutional ResNets He et al. (2016), masked autoencoders He et al. (2022)) information and 2) whether they were trained with or without supervised learning objectives.

Neural responses to the same quasi-naturalistic videos were measured in human participants using functional magnetic resonance imaging (fMRI, Hanke et al. (2016)). The human visual system includes regions showing selectivity for faces, bodies, scenes, and artifacts (Kanwisher et al., 2002; Epstein and Kanwisher, 1998; Chao et al., 1999; Downing et al., 2001). These regions were identified using independent data (a "functional localizer"), to then study their responses during the videos. The correspondence between neural representations in different regions and representations in the models was determined by calculating the correlation between their representational dissimilarity matrices (RDMs, see supplementary materials).

### 3.1.1 STATIC AND DYNAMIC INFORMATION IN CNNs AND THE BRAIN

Functional MRI responses recorded during the observation of naturalistic videos were compared to the representations in feed-forward convolutional neural networks. The same dataset (HAA-500) was used to train the different branches of a hidden-two-stream network: the "supervised static" branch (a ResNet that takes as input individual frames of a video and computes as output the action category), the "unsupervised dynamic" branch (a convolutional network trained to compute optic flow by minimizing a self-supervised loss), and the "supervised dynamic" branch (a ResNet that takes as input optic flow and computes as output the action category). In addition, to facilitate parallels with prior work, we compared neural responses to a widely studied feed-forward model: a ResNet trained with Image-net (Figure 1).

Comparing deep network models trained with the same dataset (HAA-500) showed that the self-supervised dynamic model containing optic flow information correlated with neural responses more than the supervised static model in fSTS and OFA from the face-selective network, EBA from the body-selective network, and all regions of the scene-selective network—PPA, RSP, and TOS (Fisher-transformed t-test with Bonferroni-corrected threshold). A supervised learning objective (in the

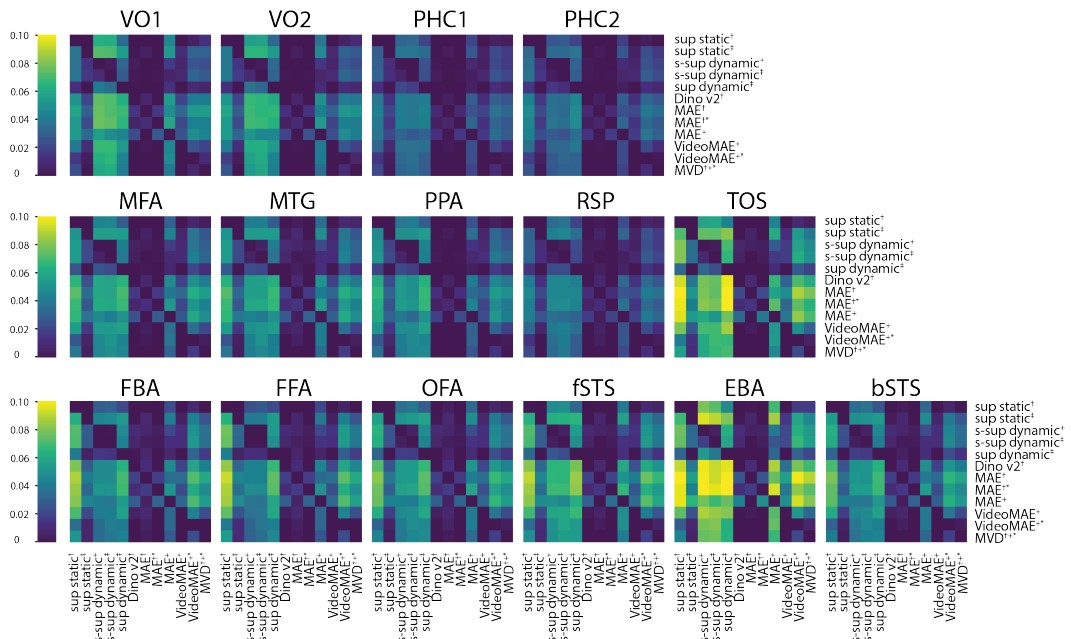

Figure 2: Models unique similarity with brain regions. The similarity was calculated using Pearson's correlation between the actual RDM of a brain region and the RDM predicted by a target model while controlling for the variation explained by a control model in the brain region. Correlations were averaged across participants. Each row corresponds to a control model and each column to a target model used for neural RDM prediction. (sup: supervised, s-sup: self-supervised, †: Image-net-trained, ‡: HAA-500-trained, +: Kinetics-400-trained, *: fine-tuned; MVD was trained on pre-trained MAE (Image-net) and VideoMAE (Kinetics-400))

supervised dynamic model) improved the similarity of the dynamic model in FFA and TOS (Fisher-transformed t-test with Bonferroni-corrected threshold)

ResNets trained with Image-Net performed well (Figure 1, first bar from the left), achieving correspondence with neural responses that in some cases surpassed that of HAA-trained models. Analyzing the differences between distinct brain regions revealed variation in the relative performance of Image-Net trained ResNets and optic-flow-based models trained on HAA. For example, responses in the extrastriate body area (EBA) were predicted equally well by dynamic models trained with HAA as well as static models trained with ImageNet, whereas responses in the fusiform body area (FBA) showed greater correspondence with the supervised static model. When the training dataset was held constant (HAA), dynamic models outperformed static models across all regions. In summary, the use of dynamic vs static information and the choice of the training dataset both affected the correspondence between models and neural representations.

Static models trained with ImageNet and of dynamic models trained on HAA achieved similar correspondence with neural responses. Therefore, we sought to determine the extent to which they accounted for unique or overlapping variance in neural responses (in section 3.2).

### 3.1.2 STATIC AND DYNAMIC INFORMATION IN MAEs AND THE BRAIN

Masked Autoencoders (MAE, He et al. (2022)) and Video Masked Autoencoders (VideoMAE, Tong et al. (2022); Feichtenhofer et al. (2022)) models are trained to reconstruct masked pixels of input (image or video) during pre-training and are further fine-tuned to predict object/action labels. MAE and VideoMAE models are very effective in learning visual representations and have been shown to outperform competing models in several visual tasks He et al. (2022); Tong et al. (2022); Feichtenhofer et al. (2022); Wang et al. (2023a); Venkatesh et al.. However, it is still unknown whether the representations learned by models based on masked autoencoding are similar to visual representations in the human brain. Here we investigated this question, quantifying the correlation between

neural responses measured with fMRI while participants watched naturalistic videos, and representations learned by models trained with masked autoencoding.

We compared the correspondence between neural responses and MAEs trained with images (which learn spatial relationships between component of an image, Wang et al. (2023b)) as well as Video-MAEs (which learn temporal relationships in videos, Wang et al. (2023b)). Finally, we also compared neural responses to masked video distillation (MVD, Wang et al. (2023b)), which combines image MAEs and videoMAEs to better capture both spatial and temporal relationships. Unlike MAE and VideoMAE, the MVD model does not aim to reconstruct missing patches at the level of pixel values. Instead, MVD adopts a knowledge-distillation approach, reconstructing missing information at the level of features extracted from pre-trained MAE and VideoMAE teachers.

As in the case of supervised models trained with the HAA dataset, models that included dynamic information (VideoMAEs) outperformed models using only static information (Image MAEs). This pattern was observed across all category-selective regions. Image MAEs did not correlate well with neural responses, even compared to HAA-trained supervised models trained with static inputs. Additionally, object identity information (in fine-tuned MAE) did not improve correlation with neural responses. Overall, the representations learned by Image MAEs were very different from neural representations. By contrast, VideoMAEs showed greater correspondence with neural responses. In particular, fine-tuning with an action recognition task (Figure 1, VideoMAE fine-tuned) improved the correspondence between VideoMAE representations and neural representations across all streams (Fisher-transformed t-values with Bonferroni-corrected threshold). Across all the pre-trained models, pre-trained MVD showed the highest similarity to neural representations in all brain streams. Further, MVD showed comparable similarity with brain streams to that of fine-tuned VideoMAE.

To further expand our investigation into the correspondence between neural representations and representations in vision Transformers, we additionally compared neural RDMs to the RDMs obtained with Dino v2, a self-supervised vision Transformer trained with a different self-supervised objective. The results revealed that worse alignment with neural responses was not restricted to Image MAEs, but extended to the Dino v2 model as well. This suggests that multiple types of self-supervised vision Transformers do not provide high correspondence with neural responses. More research will be needed to determine whether this result is due to the Transformer architecture itself.

## 3.2 DIFFERENT MODELS CAPTURE SHARED AND UNIQUE VARIANCE IN NEURAL RESPONSES

The results described in 3.1 show that representations from models trained with dynamic information are more correlated with neural representations compared to representations from models trained with static information. This overall pattern is broken by the exception of ResNets trained with ImageNet, which performed on par with models trained with supervised objective on dynamic information. This raises the question of whether ResNets trained with ImageNet and dynamic models explain overlapping variance in neural responses or whether, instead, they are complementary, capturing non-overlapping portions of the variance. This question can be posed more generally for any pair of models studied in section 3.1. We investigated this by measuring the correspondence between a "target" model's representations and the representations in each brain region while controlling for the representations encoded in a "control" model. To this end, we predicted neural representations using the representations of the control model and obtained the residuals. Then, we predicted the residuals using the representations in the target model (see supplementary materials for details).

Figure 2 demonstrates the correspondence between a target model's features and each brain region when we controlled for the features of a control model in the region's neural responses. The results are visualized as a matrix in which each row corresponds to a control model and each column to a target model. The first row of a matrix displays the correlations between models and neural responses after controlling for the Image-net-trained static model. The high values for the columns corresponding to the self-supervised dynamic and the supervised dynamic models indicate that these models and the Image-net-trained static model capture non-overlapping variance in neural responses. Representations learned by the HAA-trained self-supervised and supervised dynamic model also capture non-overlapping variance with those learned by the masked autoencoder self-supervised dynamic models: the VideoMAEs. This finding shows that despite VideoMAEs exhibit relatively high corre-

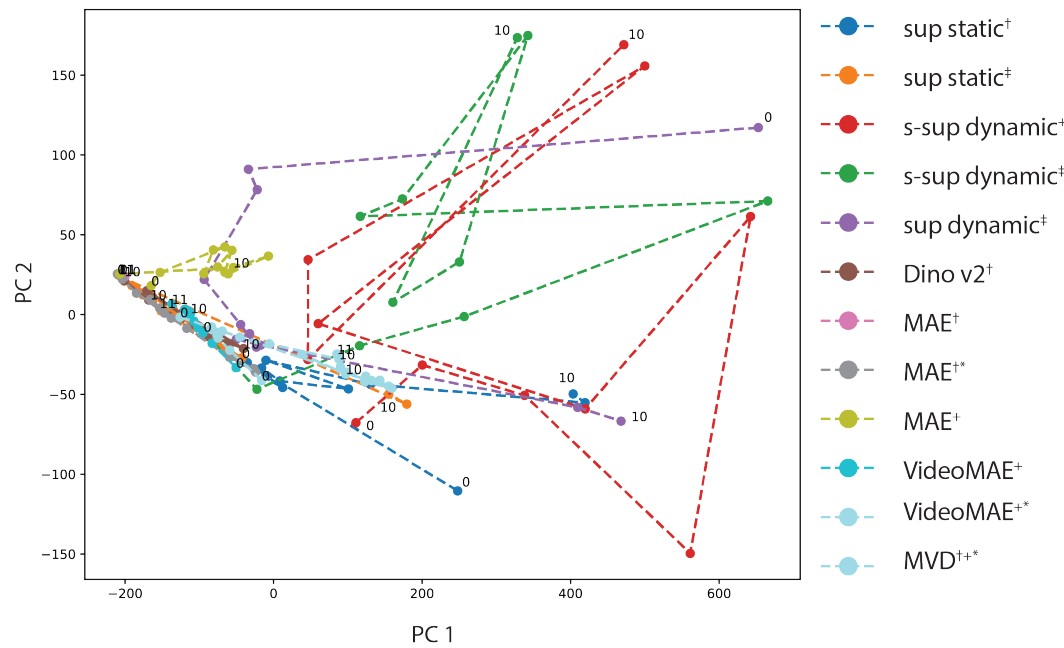

Figure 3: Principal Components (PC) of model layers' RDMs. PCs were extracted using all the layer RDMs of all the models. Each dot displays the corresponding model's layer RDM in the 2-dimensional space of PCs. Numbers on the dot (0 and 10) show the corresponding model layer number.

lations with neural responses (outperforming Image MAEs), they nonetheless fail to capture some variance in human visual representations that is accounted for by self-supervised and supervised dynamic models. Importantly, the self-supervised dynamic model accounts for unique neural variance compared to the Video MAEs even when trained on the same dataset: Kinetics (Figure2 matrices, column 3, rows 10-12). This indicates that the difference in performance between s-sup dynamic and Video MAEs cannot be fully attributed to differences in the visual diet.

VideoMAEs and MVD accounted for additional variance in neural responses compared to MAEs (as expected given the results in Figure 1) but also compared to the HAA-trained static and self-supervised dynamic models. However, they accounted for a minimal amount (if any) of additional variance compared to the supervised dynamic model, suggesting some degree of convergence on common representations across models trained with different learning objectives.

The additional unique variance explained by the optic flow models (s-sup dynamic and sup dynamic) varied across regions, being strongest in EBA and TOS and weakest in FFA. The effect was observed widely, in regions previously associated with the processing of dynamic information (such as STS), but also in ventral temporal regions that have not been typically associated with the representation of dynamics (such as PPA). This observation is consistent with recent work suggesting that dynamic information is represented in a broader range of brain regions than previously thought (Robert et al. (2023)).

As a key takeaway, the results show that the CNN models using optic flow (namely, s-sup dynamic and sup dynamic) explain unique neural variance that is not captured by MAE models (Figure 2, columns 3-5 and rows 7-12 of the matrices). Importantly, they also explain unique variance that is not captured by other CNNs – even when they are trained with the same dataset (HAA, Figure 2, columns 4-5, row 2 of the matrices). The results therefore indicate that the difference between CNN and Transformer architectures alone is not sufficient to account for the unique variance in neural responses explained by the models using optic flow.

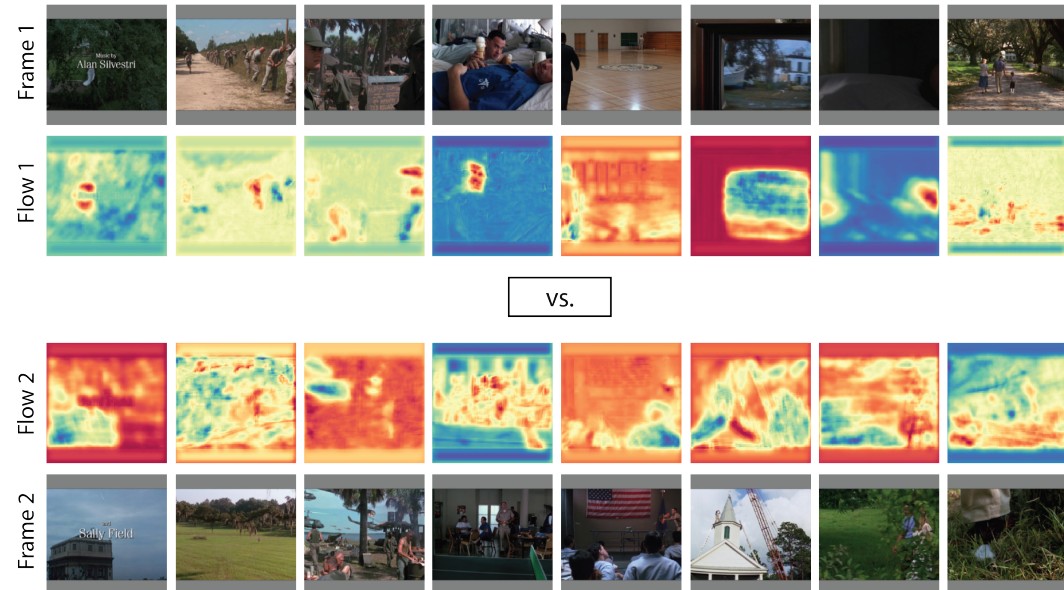

Figure 4: Visualization of pairs of frames with very different loadings along the second principal component in the space of the models' representational dissimilarity matrices. Each column illustrates the frames' appearance and their optic flow. Images with different loadings along the second principal component typically show large differences in the overall amount of optic flow.

### 3.3 MODELS WITH OPTIC FLOW INFORMATION BETTER CONVERGE WITH NEURAL RESPONSES

To better understand the difference between the representational pattern of models, we extracted components using the Principle Component Analysis (PCA) algorithm that best captures variation in the RDMs of all layers of all models across two dimensions. Figure 3 demonstrates the trajectory of layer-to-layer change in the representational pattern of each model across two PCs. The variation in the layers' RDMs of HAA-trained dynamic models that process optic-flow information is largely captured with the second PC.

In other words, the results of principal component analysis (Figure 3) reveal that layers in the models using optic flow representations encode representations with a fundamentally different representational geometry compared to the other models. This is evidenced by the higher loadings of the optic flow models on the second principal component. By contrast, layers in the MAE models as well as in the CNNs that do not use optic flow information have lower loadings on the second principal component.

Figure 4 displays eight example pairs of timepoints showing high degree of dissimilarity along the second principal component. The timepoints in the pairs vary substantially in terms of the overall amount of optic flow present at the two timepoints.

## 4 CONCLUSION

Despite the effectiveness of MAEs at several vision tasks, their correspondence with neural responses was relatively low compared to convolutional neural networks, making MAEs an exception to the previously observed correlation between a model's categorization performance and its ability to account for neural responses Yamins et al. (2014). Video MAEs substantially outperformed image MAEs in their correspondence to human representations. Similarly, convolutional models using optic flow outperformed convolutional models based on static features, highlighting the importance of dynamic information for human visual representations. This phenomenon was observed even in brain regions traditionally associated with the processing of static information, in line with recent work showing that these regions also respond to dynamic stimuli (Robert et al., 2023). In future

work, it will be important to enrich the analyses by comparing neural responses to models using additional metrics, such as Brain Score Schrimpf et al. (2018).

Convolutional models based on optic flow explained unique variance in neural responses that was not accounted for by any other model, not even video MAEs. Analysis of the representational geometry in the different layers of the models revealed that the second principal component in the space of representational dissimilarity matrices (RDMs) distinguished between convolutional models based on optic flow on one hand (which scored highly on the component) and all the other models on the other hand, suggesting a critical role of optic flow representations in human vision. We probed this conclusion further by examining the loadings of this component, and identifying pairs of scenes in the movie that were differentiated by the models based on optic flow but not by the other models. These included scenes with similar entities and backgrounds, that differed in the presence or absence of overall background flow (e.g. due to movement of the camera), further supporting the conclusion that video MAEs do not encode a set of dynamic features that are instead computed by both optic flow models and by human vision.

The difference in alignment with neural responses between MAEs and CNNs is likely also driven in part by additional factors above and beyond optic flow. In particular, the comparison between Im­ageMAEs and the static net trained with ImageNet indicates that differences in architecture and task also play an important role for the differences in alignment with neural responses. In conclusion, the results converge to indicate that the lack of optic flow representations and the use of self-supervised Vision Transformer architectures are jointly responsible to account for decreased alignment between models and neural representations.

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

## A    SUPPLEMENTARY MATERIALS

### A.1    DATA

BOLD fMRI responses (3×3×3 mm) to eight movie segments of 'Forrest Gump' were obtained from the publicly available *studyforrest* audiovisual dataset (`http://studyforrest.org`). Fifteen right-handed participants took part in the study (6 females; age range 21-39 years, mean 29.4 years). The data was acquired with a T2*-weighted echo-planar imaging sequence, using a whole-body 3 Tesla Philips Achieva dStream MRI scanner equipped with a 32 channel head coil.

### A.2    PREPROCESSING

Data were first preprocessed using fMRIPrep (`https://fmriprep.readthedocs.io/en/latest/index.html`): a robust pipeline for the preprocessing of diverse fMRI data. Anatomical images were skull-stripped with ANTs (`http://stnava.github.io/ANTs/`), and FSL FAST was used for tissue segmentation. Functional images were corrected for head movement with FSL MCFLIRT (`https://fsl.fmrib.ox.ac.uk/fsl/fslwiki/MCFLIRT`), and were subsequently coregistered to their anatomical scan with FSL FLIRT. Finally, the skull-stripped anatomical images were normalized to the MNI template using SPM. We denoised the data with CompCor Behzadi et al. (2007) using 5 principal components extracted from the union of cerebrospinal fluid and white matter.

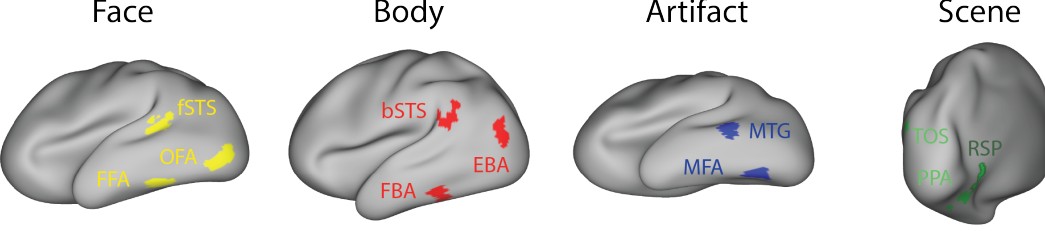

Figure 5: Masks of visual category-selective regions in the human brain projected on an inflated cortical surface in MNI space.

### A.3    REGIONS OF INTEREST (ROI)

We used the first block-design run from the category localizer session, to identify four sets of category-selective brain regions: face-selective areas (occipital face area - OFA, fusiform face area - FFA, face-selective posterior superior temporal sulculs - fSTS), body-selective areas (extrastriate body area - EBA, fusiform body area - FBA, body-selective posterior superior temporal sulcus - bSTS), artifact-selective areas (medial fusiform gyrus - MFG, medial temporal gyrus MTG), and scene-selective areas (transverse occipital sulcus - TOS, parahippocampal place area - PPA, retrosplenial cortex - RSC) 5. The analysis was conducted using a standard GLM with FSL FEAT Woolrich et al. (2001), where each seed ROI was defined as a sphere with a 9mm radius centered on the peak of the corresponding contrast (e.g., faces > bodies, objects, scenes, and scrambled images for face-selective regions). For each ROI, we combined data from the left and right hemispheres and selected the 80 voxels with the highest t-values for the preferred category compared to other categories.

To identify the visual regions in ventral and dorsal brain streams, we used an atlas of probabilistic maps of visual topography in the human cortex from a previous study Wang et al. (2015). A list of probabilities is associated with each voxel to reflect the likelihood of that voxel being part of each

of the brain regions. We calculated the transformation from MNI space to each participant's native space and co-registered the probability maps with each participant's anatomy. To prevent overlap between the regions of interest in the participant's native space, we followed a procedure analogous to Wang et al. (2015). Specifically, we calculated the maximum probability map for each participant, using which we exclusively classified each voxel as either belonging to a specific ROI or as being outside of all the ROIs. Eleven visual regions were included in the experiments: ventral and dorsal V1 (V1v, V1d), V2 (V2v, V2d), V3 (V3v, V3d), in addition to V4, posterior and anterior ventral occipital (VO1, VO2), and parahippocampal areas (PHC1, PHC2).

## A.4    MODELS' REPRESENTATIONAL DISSIMILARITY MATRICES (RDM)

In order to compare the models and the fMRI data, we computed representational dissimilarity matrices (RDMs) for the models' layers with a multi-step procedure. First, since the temporal resolution of the models' representations (25Hz) is much higher than the temporal resolution of fMRI data, we down-sampled each layer's activation timecourses over time by selecting one data point every five time points(down to 5 Hz). Then, we convolved the layer's activations with a standard Hemodynamic Response Function (HRF). Given that the fMRI data's repetition time (TR) is 2 seconds, we took a layer's activation every $25 \times 2 = 50$ time points.

Finally, for each layer we computed the dissimilarities between all pairs of timepoints, obtaining RDMs in which the entry at column $j$ and row $i$ contains correlation dissimilarity (1-Pearson's r) between the layer activations at time $i$ and time $j$. We repeated this procedure for BOLD responses to all eight movie segments, resulting in eight RDMs.

## A.5    BRAIN REPRESENTATIONAL DISSIMILARITY MATRICES (RDM)

RDMs were constructed separately for each ROI in the subject's native space. For each region, we calculated the correlation dissimilarity ($1 - r$ where r is Pearson's correlation) of fMRI response patterns for all pairs of TRs. This yielded eight RDMs, corresponding to BOLD responses in eight video segments.

## A.6    MEASURING MODELS SIMILARITY WITH BRAIN DATA

To evaluate how well each model accounts for the activity in the ROIs, we used a cross-validated linear regression to predict the left-out movie segment brain region RDM and computed the correlation between the predicted and the true RDM in each brain region. The correlation captures how well a model's layers can predict a brain region's responses to visual stimuli. First, we used each model's layers' RDMs corresponding to seven (out of eight) video segments to train a linear regression model that predicts the corresponding seven RDMs in each brain region. Then, we averaged the linear regression model's coefficients along the seven segments and used the averaged coefficients to predict the brain region RDM of the left-out segment, using the model layers' RDMs of the corresponding segment. Finally, we calculated the Pearson's correlation between the predicted and the true RDMs. We repeated the leave-one-out cross-validation process for all the segments and averaged over the obtained correlations.

## A.7    MEASURING UNIQUE AND SHARED SIMILARITY OF A PAIR OF MODELS WITH BRAIN DATA

To evaluate how well unique and shared features among a pair of computational models correspond to the brain data, we used Pearson's r to measure the accuracy of a "target" model's layers prediction of a brain region RDM while controlling for the variation of a "control" model layers. Using leave-one-out cross-validation, first, we estimated the coefficients of a linear regression model that predicts a brain region's RDM from the control model's layers in training video segments (seven out of eight). Second, we subtracted the predicted from the actual brain region RDM in the training and the left-out video segments to obtain training and left-out residuals. Third, we estimated the coefficients of a linear regression model that predicts training residuals of each video segment using the target model layers. Finally, we measured Pearson's correlation between the target model's prediction of the left-out video segment residuals and the residuals obtained from the prediction of the control model.

Table 2: Test performance of models on the HAA500 dataset

| Model | epochs | Performance | |
| --- | --- | --- | --- |
| | | Top-1 | Top-3 |
| sup static | 47 | 30.80% | 49.38% |
| unsup dynamic + sup dynamic | 12, 50 | 22.72% | 37.90% |

### A.7.1 TRAINING AND TESTING THE TWO-STREAM CNN FOR ACTION RECOGNITION

We adopted the models in Zhu et al. (2019) and trained on the HAA500 dataset Chung et al. (2021). The dataset contains over 591k labeled frames with 500 action classes. 85% of the data points were used for training, 5% for validation, and 10% for testing A.7.1. The training dataset was converted to the Webdataset format, i.e., shards of tar files. We used 4 V100 GPUs and 8 workers to load the dataset and train the models. All the analyses were performed on the same version of the movie that was used to acquire fMRI responses in the StudyForrest dataset Hanke et al. (2016).

The *supervised static model* have a ResNet18 architecture He et al. (2016), and were trained for 47 epochs with a batch size of 128. The training was done with the stochastic gradient descent algorithm with a 0.001 initial learning rate and a 0.0001 weight decay. During training, the gradients were accumulated and backpropagated for every two batches. Each frame in an input batch is a $224 \times 224$ frame and was randomly flipped horizontally.

The *unsupervised dynamic model* was trained for 12 epochs with a batch size of 32 and an initial learning rate of 0.01. No weight decay was used during training. Input to this model consists of a set of 11 frames each with dimensions of $224 \times 224$.

The *supervised dynamic model* was trained for 50 epochs with a batch size of 128 and an initial learning rate of 0.001. A weight decay of 0.0005 was used to train the models, and the gradients were accumulated and backpropagated every 5 batches.

