# OpenReview forum: "Visual Representations in Humans and Machines: A Comparative Analysis of Artificial and Biological Neural Responses to Naturalistic Dynamic Visual Stimuli"
_ICLR.cc/2025/Conference — Submitted to ICLR 2025_

### Official Review · Reviewer_Tz3t · 2024-11-03

**Soundness:** 3
**Presentation:** 3
**Contribution:** 3
**Rating:** 5
**Confidence:** 4

**Summary:**

This paper compares the alignment of the response patterns in a variety of MAE and CNN, trained with static or dynamic stimuli, with the fMRI response patterns in the different visual areas of the human cortex. They found that (1) MAE is not doing better than CNN,  (2) networks trained with dynamic stimuli are more aligned than one trained with static stimuli, and (3) dynamic and static models can be complementary, i.e., combining them together can yield better alignment.

**Strengths:**

The study is reasonably motivated, and the questions are well-posed.
Comparing the alignment of CNN and MAE trained with static and dynamic stimuli with human systems is a legitimate problem.
Systematic evaluation of pairwise combinations of different models and finding some complementary contributions is interesting.
The paper is relatively well-written and clear.
The approach and the methodology are straightforward. Some of the findings are interesting.

**Weaknesses:**

The approach is straightforward. The conceptual and technical advance is limited.
The insights provided by the paper are rather limited.
It is a good paper with some interesting results, but probably on par with the standard of ICLR papers.

**Questions:**

It would be worthwhile to dig deeper to understand WHY some models are better than others and why some models are complementary.

---

> ### Author Response · Authors · 2024-11-26
> **Response to Reviewer Tz3t**
>
> >> Weaknesses:
> >> The approach is straightforward. The conceptual and technical advance is limited. The insights provided by the paper are rather limited. It is a good paper with some interesting results, but probably on par with the standard of ICLR papers.
>
> We appreciate the Reviewer’s evaluation that the paper is on par with the standard of ICLR papers. In order to improve the paper in the direction indicated by the Reviewer’s questions, we have expanded the analyses, clarified their rationale, and discussed in more detail the interpretation of the results.
>
> >> Questions:
> >> It would be worthwhile to dig deeper to understand WHY some models are better than others and why some models are complementary.
>
> We agree with Reviewer Tz3t that a central question concerns why some of the models – and specifically the s-sup dynamic and the sup dynamic models – explain unique neural variance that is not explained by the other models. One possible hypothesis is that these models outperform the other models due to their CNN architecture, that differentiates them from the MAE models which are based on a Transformer architecture. However, if this were the case, the sup static model trained with the HAA dataset should perform equally well (because it is also based on a CNN architecture). However, this is not the case. This suggests that the use of CNN architecture is not sufficient to explain why the s-sup dynamic and the sup dynamic models explain unique neural variance.
>
> Another possibility is that the difference might be driven by the visual diet: the s-sup dynamic and the sup dynamic models were trained with the HAA dataset, while other models (e.g. the video MAEs) were trained using the Kinetics dataset. To rule this out, we added new analyses with a s-sup dynamic model trained with Kinetics. The new analyses show that even when trained on the same dataset as the Video MAEs (Kinetics), the s-sup model still explains unique neural variance that is not accounted for by the Video MAEs, suggesting that differences in the visual diet are not sufficient to explain why the s-sup dynamic and sup dynamic models explain unique neural variance that is not accounted for by the Video MAEs.
>
> Finally, an alternative hypothesis is that the s-sup dynamic and the sup dynamic models differ from all other models in terms of their representation of optic flow information. Figure 4 shows that these models differ from all other models along the second principal component in representational space. However, Figure 4 alone does not tell us whether this second principal component is related to optic flow information. For this reason, we performed the analysis shown in Figure 5. We identified pairs of timepoints in the video that show large differences along principal component 2, and compared them. The comparison reveals that timepoints that differ along principal component 2 are very different in terms of the optic flow present at those timepoints. In other words, this result indicates that principal component 2 is associated with differences in optic flow, and thus ultimately that the models that account for unique neural variance (s-sup dynamic and sup dynamic) differ from the rest of the models in terms of their optic flow representations.
>
> We have edited the manuscript to explain these results more clearly, adding the following passages to the Results section:
>
> “As a key takeaway, the results show that the CNN models using optic flow (namely, s-sup dynamic and sup dynamic) explain unique neural variance that is not captured by MAE models (Figure 3, columns 3-5 and rows 5-9 of the matrices). Importantly, they also explain unique variance that is not captured by other CNNs – even when they are trained with the same dataset (HAA, Figure 3, columns 3-4, row 2 of the matrices). The results therefore indicate that the difference between CNN and Transformer architectures alone is not sufficient to account for the unique variance in neural responses explained by the models using optic flow.”
>
> And
>
> “Importantly, the self-supervised dynamic model accounts for unique neural variance compared to the Video MAEs even when trained on the same dataset: Kinetics (Figure 3 matrices, column 3, rows 10-12). This indicates that the difference in performance between s-sup models and Video MAEs cannot be fully attributed to differences in the visual diet.”

---

> > ### Author Response · Authors · 2024-11-26
> > **Response to Reviewer Tz3t - continued**
> >
> > We have also added analyses with another Transformer architecture, trained with a different learning objective: Dino v2. We have added the results to the figures, as well as the following text:
> >
> > “To further expand our investigation into the correspondence between neural representations and representations in vision Transformers, we additionally compared neural RDMs to the RDMs obtained with Dino v2, a self-supervised vision Transformer trained with a different self-supervised objective. The results revealed that worse alignment with neural responses was not restricted to Image MAEs, but extended to the Dino v2 model as well. This suggests that multiple types of self-supervised vision Transformers do not provide high correspondence with neural responses. More research will be needed to determine whether this result is due to the Transformer architecture itself.”
> >
> > To investigate more closely why some models explain unique neural variance, we treated the representational dissimilarity matrix of each model’s layer as a high-dimensional vector. Analyzing all models and layers, we obtained a set of vectors, and performed principal component analysis (PCA) to identify dimensions that account for most variation in the representations encoded in different layers of different models (Figure 4). We found that the models that use optic flow learn very different representational structure compared to both MAE models and other CNNs (see the green and red lines). This observation converges with the argument outlined in the previous paragraph to indicate that the differences are not due to an overall distinction between CNN and Transformer architectures. Instead, among CNNs, those that use optic flow learn different representations compared to all other models, including video MAEs.
> > To clarify the significance of Figure 4, and how it speaks to the question of why some models explain unique neural variance, we added the following passage:
> >
> > “In other words, the results of principal component analysis (Figure 4) reveal that layers in the models using optic flow representations encode representations with a fundamentally different representational geometry compared to the other models. This is evidenced by the higher loadings of the optic flow models on the second principal component. By contrast, layers in the MAE models as well as in the CNNs that do not use optic flow information have lower loadings on the second principal component.”
> >
> > At this point, we wanted to obtain a more interpretable understanding of the stimulus features driving the differences between the models that use optic flow and the rest of the models. We noted that the models that use optic flow mostly differ from the rest of the model along the second principal component (PC2). Principal components have loadings on each element of the representational dissimilarity matrix. Since the representational dissimilarity matrix contains the dissimilarity between different timepoints in the video, we could identify which pairs of timepoints in the video have high loadings for PC2. We visualized additional example pairs of timepoints with high loadings for PC2 in Figure 5, showing the frames as well as optic flow maps at those timepoints. Typically, one timepoint in the pair is characterized by more optic flow than the other, further supporting the hypothesis that the differences observed between the models are driven by how they represent optic flow information.
> >
> > Finally, we have clarified our interpretation of the overall pattern of results, stating more clearly our interpretation of the results in the Conclusion section:
> >
> > “In conclusion, three models in particular explained unique variance in the neural responses that was not captured by other models: s-sup dynamic trained with Kinetics, s-sup dynamic trained with HAA and sup dynamic trained with HAA. These are the three models that represent optic flow information. The results converge to indicate that the use of optic flow information is a key factor that differentiates these models from the other models tested.”
> >
> > We apologize for the lack of clarity in the original version of the paper. We hope that Reviewer Tz3t will consider updating the score in light of the revisions.

---

> > > ### Author Response · Authors · 2024-12-02
> > > **Follow up**
> > >
> > > As today is the last day that reviewers may post a message to the authors, we welcome any acknowledgment of our revisions and any additional feedback. Thank you.

---

> > > > ### Comment · Reviewer_Tz3t · 2024-12-03
> > > > **Thanks for the valiant effort**
> > > >
> > > > I have read all the responses and the various backs and forth between the authors and the other reviewers.  A lot of information has been added relative to the original submission.  It is simply a bit too much for me to digest, even though I applauded the authors' valiant effort.  While I do find the findings promising, the presentation leaves much to be desired. A clear articulation of the theory is really missing, particularly in the original submission. I have the same sentiment as reviewer kFNo, and I am maintaining my score.

---

> > > > > ### Author Response · Authors · 2024-12-03
> > > > > **Response to Reviewer Tz3t - round 2**
> > > > >
> > > > > We thank Reviewer Tz3t for their feedback, and we understand that it takes time to evaluate the additional work. We note that reviewer kFNo has revised their score upwards in light of the revisions.

---

### Official Review · Reviewer_Xyvz · 2024-11-04

**Soundness:** 3
**Presentation:** 2
**Contribution:** 2
**Rating:** 6
**Confidence:** 3

**Summary:**

The authors compare the representation layer of image and video MAEs to human fMRI response data for a variety of brain areas, and find that the models tested have dissimilar representations compared to human responses. The paper reports fine-tuning videoMAEs on an action recognition task, which results in improved fMRI response prediction, but does not achieve the predictive power of a convolutional network. The authors show that combining image and video MAE together gets closer, as does models trained on optic flow rather than movies.

**Strengths:**

Originality: The paper is, to my knowledge, novel in evaluating MAEs, supervised vs self-supervised and image vs video networks using human neural data such as fMRI, and addresses a gap in the literature

Quality: The reported experiments appear to be well-designed, and the findings appear credible.

Clarity: The individual figure plots are clear in terms of the metric being evaluated for different brain regions.

Significance: Investigating the representational alignment of newer, high-performing models like MAEs with human cognition is both interesting and important. It is a surprising result that the better model performance in MAEs does not necessarily improve representational alignment with humans, and challenges the assumption that better performance would come with closer alignment.

**Weaknesses:**

Overall, the paper is confusing as written, and this results in difficulty understanding that muddies what is I believe otherwise exciting work. For example, many of the results are simply reported with little to no summarization or interpretation of the larger significance & meaning.

There is an overall lack of justification for the methods chosen, I believe due to lack of relevant citations to previous work. For example, I could not find a justification for the author’s choice of comparison metric for model/RDM. The use of pearson’s correlation appears sound, but seems limited, especially because this requires training a linear predictor on the output, reducing the amount of test data to 1/7 of the movie segments tested. This method needs better justification citing previous work that has established this metric, or ideally an additional metric to validate it.

The PCA analysis in particular needs to be explained and justified more clearly. This is true for figure 4 but especially the analysis in figure 5 is not well explained, and the one shown appears to be an illustrative example.

**Questions:**

Why do the authors think that ViT MAEs, despite improved performance over CNNs, do not show strong representational alignment? Some discussion of this would help strengthen the paper.

Relatedly, more summarization of the results throughout the paper would make this easier to read.

How do the authors account for differences in temporal resolution between these frame-by-frame models, which operate on videos with <100ms frame rate, and the much lower temporal resolution of fMRI?

To improve clarity and ease of summarization, the authors should have more visualizations that summarize data effectively. For example, figure 1 could have an additional plot that compares all brain areas together, normalized by the noise ceiling and/or the explainability of the best model (CNN).

Because F6 is in the supplemental, F1/2/3 should label Face/Body/Artifact/Scene on the left side rows to make this designation clear.

Give more justification for the PCA section, especially figures 4 and 5. I have read through this section and figures multiple times and am still unclear why these analyses are helpful in understanding the differences between image/video/optic flow model representations, I especially find figure 5 confusing - why this is useful to visualize in terms of improving understanding of representational alignment to human fMRI data? Was this analysis performed over multiple frames and can it be summarized?

Do the authors control for model size when comparing representational similarity? If not, this seems problematic, as one could expect a larger model would perform better, especially given the chosen method of using a trained linear predictor on the model outputs to compare to RDMs.

It would be interesting to see how the models tested perform on brain score - another independent measure of biological similarity - it would be interesting to see if the same trends for MAEs are seen for this measure.

Add figure references in section 3.1.1

Fix Formatting/spelling/grammar issues:
038: ‘...similarly flexible representations: Vision Foundation Models’
064: Opic flows
117:  learn image representations
Figure 2: there is no a), b) part of the figure.
468/469: c, but also by the pc.

---

> ### Author Response · Authors · 2024-11-27
> **Response to Reviewer Xyvz**
>
> >>> Weaknesses:
> >>> Overall, the paper is confusing as written, and this results in difficulty understanding that muddies what is I believe otherwise exciting work. For example, many of the results are simply reported with little to no summarization or interpretation of the larger significance & meaning.
> >>> There is an overall lack of justification for the methods chosen, I believe due to lack of relevant citations to previous work. For example, I could not find a justification for the author’s choice of comparison metric for model/RDM. The use of pearson’s correlation appears sound, but seems limited, especially because this requires training a linear predictor on the output, reducing the amount of test data to 1/7 of the movie segments tested. This method needs better justification citing previous work that has established this metric, or ideally an additional metric to validate it.
> >>> The PCA analysis in particular needs to be explained and justified more clearly. This is true for figure 4 but especially the analysis in figure 5 is not well explained, and the one shown appears to be an illustrative example.
>
> We fully agree with the limitations identified by Reviewer Xyvz, we have edited the manuscript extensively to address the lack of clarity regarding the choice of models and methods and to provide more insight regarding the interpretation of the results.
>
>
>
> >>> Questions:
> >>> Why do the authors think that ViT MAEs, despite improved performance over CNNs, do not show strong representational alignment? Some discussion of this would help strengthen the paper.
>
> This is an excellent question. We hypothesize that this is due to a convergence between two factors. First, we found that two models explain unique neural variance compared to most other models: s-sup dynamic and sup dynamic. They are the only models trained to use optic flow information. They explain unique neural variance compared to MAEs, and also compared to other models that have analogous architecture (the static CNNs trained on the same dataset). We have added a new model to the analysis, a s-sup dynamic model trained on Kinetics (so that it is trained with the same visual diet as the Video MAEs). In line with our interpretation of the results, the Kinetics-trained s-sup dynamic model accounted for unique neural variance compared to the Video MAEs despite having been trained on the same dataset. We have summarized this key point as follows:
>
> “As a key takeaway, the results show that the CNN models using optic flow (namely, s-sup dynamic and sup dynamic) explain unique neural variance that is not captured by MAE models (Figure 2b, columns 3-5 and rows 7-12 of the matrices). Importantly, they also explain unique variance that is not captured by other CNNs – even when they are trained with the same dataset (HAA, Figure 2b, columns 4-5, row 2 of the matrices). The results therefore indicate that the difference between CNN and Transformer architectures alone is not sufficient to account for the unique variance in neural responses explained by the models using optic flow.”
>
> Importantly, additional factors beyond optic flow are likely to contribute to the difference in representational alignment of ViT MAEs. This is highlighted in particular by the fact that the static sup CNN trained with ImageNet shows substantially higher representational alignment with neural responses compared to ViT MAEs. In order to further examine whether this phenomenon is limited to MAEs or whether it extends to other self-supervised vision transformers as well, we have included Dyno v2 in the analysis –  a Vision Transformer trained with a different self-supervised learning objective. Dyno v2 also showed generally poor alignment with neural responses, showing that the worse alignment is not restricted exclusively to Vision Transformers trained with masked autoencoding.
>
> “It is important to note however that the representation of optic flow is likely not the only contributor to the worse alignment of the MAEs. In fact, even when trained with an action dataset (HAA), the static CNN showed better alignment with neural responses compared to the Image MAEs, and accounted for unique neural variance that was not captured by the MAEs.”

---

> ### Author Response · Authors · 2024-11-27
> **Response to Reviewer Xyvz - part 2**
>
> We also highlighted this in the Conclusion:
>
> “The difference in alignment with neural responses between MAEs and CNNs  is likely also driven in part by additional factors above and beyond optic flow. In particular, the comparison between ImageMAEs and the static net trained with ImageNet indicates that differences in architecture and task also play an important role for the differences in alignment with neural responses.”
>
>
> >>> Relatedly, more summarization of the results throughout the paper would make this easier to read.
>
> We appreciate this suggestion, we have added a brief summary of the implications of each result. Note that Figures 2,3 have been combined into a single Figure (Figure 2), Figures 4 and 5 have been renamed to Figures 3 and 4. Here is a list of the edits:
>
> “As a key takeaway, the results show that the CNN models using optic flow (namely, s-sup dynamic and sup dynamic) explain unique neural variance that is not captured by MAE models (Figure 3, columns 3-5 and rows 5-9 of the matrices). Importantly, they also explain unique variance that is not captured by other CNNs – even when they are trained with the same dataset (HAA, Figure 3, columns 3-4, row 2 of the matrices). The results therefore indicate that the difference between CNN and Transformer architectures alone is not sufficient to account for the unique variance in neural responses explained by the models using optic flow.”
>
> and
>
> “In other words, the results of principal component analysis (Figure 3) reveal that layers in the models using optic flow representations encode representations with a fundamentally different representational geometry compared to the other models. This is evidenced by the higher loadings of the optic flow models on the second principal component. By contrast, layers in the MAE models as well as in the CNNs that do not use optic flow information have lower loadings on the second principal component.”
>
> As well as
>
> “In conclusion, three models in particular explained unique variance in the neural responses that was not captured by other models: s-sup dynamic trained with Kinetics, s-sup dynamic trained with HAA and the flow classifier trained with HAA. These are the three models that represent optic flow information. The results converge to indicate that the use of optic flow information is a key factor that differentiates these models from the other models tested.”
>
> We are working on the responses to the other comments, and we will post them shortly.

---

> ### Author Response · Authors · 2024-11-27
> **Response to Reviewer Xyvz - part 3**
>
> >>> How do the authors account for differences in temporal resolution between these frame-by-frame models, which operate on videos with <100ms frame rate, and the much lower temporal resolution of fMRI?
>
> This is an excellent question. Model representations were computed at the original frame rate (25Hz), and were then downsampled to a rate of 5Hz. This yields a temporal resolution that is still higher than that of fMRI responses. Next, features were extracted from the models’ layer at each frame. Next, the features were convolved with a standard haemodynamic response function (HRF) to account for the shape of the BOLD response. Finally, the convolved features were further downsampled to the temporal resolution of fMRI data to obtain model RDMs matching the size of the neural RDMs. We hypothesized that extracting features at a higher temporal resolution and downsampling after the convolution with the HRF would yield more accurate results by preserving information about the impact of finer grained temporal variation in visual features on the BOLD responses.
>
> >>> To improve clarity and ease of summarization, the authors should have more visualizations that summarize data effectively. For example, figure 1 could have an additional plot that compares all brain areas together, normalized by the noise ceiling and/or the explainability of the best model (CNN).
>
>
> >>> Because F6 is in the supplemental, F1/2/3 should label Face/Body/Artifact/Scene on the left side rows to make this designation clear.
>
> As recommended, we have added labels to Figure 1 indicating what regions are selective for Faces/Bodies/Artifacts/Scenes. In Figures 2 and 3 (now combined into Figure 2), due to the large number of matrices we do not report the matrices of regions selective for different categories on different rows, and since preferred category does not seem to be a main driver of the effects we have omitted this information. Including a normalized version of the plots is a good idea, but since several new regions were added to Figure 1 (V1v, V1d, V2v, V2d, V3v, V3d, V4) in order to respond to other suggestions, we are short on space available to add more plots.
>
> >>> Give more justification for the PCA section, especially figures 4 and 5. I have read through this section and figures multiple times and am still unclear why these analyses are helpful in understanding the differences between image/video/optic flow model representations, I especially find figure 5 confusing - why this is useful to visualize in terms of improving understanding of representational alignment to human fMRI data? Was this analysis performed over multiple frames and can it be summarized?
>
> We apologize for the lack of clarity in the original version. Figures 4 and 5 (now renamed to 3 and 4) aimed to compare the models between each other, with the goal of identifying differences between the models that explained unique variance in neural responses (s-sup dynamic and sup dynamic) and the other models. In the original version of the manuscript, the rationale and interpretation of Figures 4 and 5 (now 3 and 4) was not adequately explained. We have addressed this limitation including the following passage in the Results section to explain the interpretation of Figure 4 (now renamed as Figure 3)
>
> “the results of principal component analysis (Figure 3) reveal that layers in the models using optic flow representations encode representations with a fundamentally different representational geometry compared to the other models. This is evidenced by the higher loadings of the optic flow models on the second principal component. By contrast, layers in the MAE models as well as in the CNNs that do not use optic flow information have lower loadings on the second principal component.”

---

> > ### Author Response · Authors · 2024-11-27
> > **Response to Reviewer Xyvz - part 4**
> >
> > Regarding Figure 5 (now renamed as Figure 4), we wanted to obtain a more interpretable understanding of the stimulus features driving the differences between the models that use optic flow and the rest of the models. We noted that the models that use optic flow mostly differ from the rest of the model along the second principal component (PC2). Principal components have loadings on each element of the representational dissimilarity matrix. Since the representational dissimilarity matrix contains the dissimilarity between different timepoints in the video, we could identify which pairs of timepoints in the video have high loadings for PC2. We expanded the figure to include 8 example pairs of timepoints with high loadings for PC2, showing the frames as well as optic flow maps at those timepoints. Typically, one timepoint in the pair is characterized by more optic flow than the other, further supporting the hypothesis that the differences observed between the models are driven by how they represent optic flow information. We added the following caption to explain the findings:
> >
> > “Visualization of pairs of frames with very different loadings along the second principal component in the space of the models’ representational dissimilarity matrices. Each row illustrates the frames’ appearance and their optic flow. Images with different loadings along the second principal component typically show large differences in the overall amount of optic flow.”
> >
> > We interpret the results as indicating that differences in optic flow representation distinguish between the models that account for unique neural variance (s-sup dynamic and sup dynamic) and the rest of the models.
> >
> >
> >
> > >>> Do the authors control for model size when comparing representational similarity? If not, this seems problematic, as one could expect a larger model would perform better, especially given the chosen method of using a trained linear predictor on the model outputs to compare to RDMs.
> >
> > This is an important question that led us to clarify some key aspects of the methods. We did not train a linear predictor on the model outputs to compare to RDMs, but rather we first computed an RDM for each model layer, and then trained a linear weighting of the RDMs for different layers. Thus, the number of parameters is the number of layers. The number of layers used were comparable across the models. However, to further mitigate concerns about the free parameters in this procedure, the weights for the layers were estimated using 7 of the 8 runs, and brain-model correspondence was computed using the left out run, with zero free parameters. We have added a section to the Methods that details this procedure, titled “Comparison Between Models and Neural Responses”.
> >
> > >>> It would be interesting to see how the models tested perform on brain score - another independent measure of biological similarity - it would be interesting to see if the same trends for MAEs are seen for this measure.
> >
> > Yes, this would be a very interesting analysis. We were not able to complete it in the time available, but we have added this passage to the manuscript to highlight that this is an important future step:
> > “In future work, it will be important to enrich the analyses by comparing neural responses to models using additional metrics, such as Brain Score \cite{schrimpf2018brain}.”
> >
> > >>> Add figure references in section 3.1.1
> >
> > Thanks, we fixed this.
> >
> > >>> Fix Formatting/spelling/grammar issues: 038: ‘...similarly flexible representations: Vision Foundation Models’ 064: Opic flows 117: learn image representations Figure 2: there is no a), b) part of the figure. 468/469: c, but also by the pc.
> >
> > We have addressed these issues.

---

> > > ### Comment · Reviewer_Xyvz · 2024-11-28
> > > **Response to Rebuttal**
> > >
> > > I appreciate the author’s updating their paper text and figures.
> > >
> > > Figure 1 is more understandable now.
> > >
> > > In addition, the summarization and interpretation of key points throughout the paper improves the readability overall.
> > >
> > > For the PCA figure, the additional text clarifications do clear up what the claims of interpretation for the figure are - however I believe these are over-stated. What is the author’s justification for the claim that the difference in loadings in PCA2 indicates a fundamentally different representational geometry in the optic flow models? Perhaps a different type of plot such as tSNE or UMAP would be more appropriate here, if the goal is to highlight differences in the underlying manifold structures. Regardless, any such claims would need to be substantiated. Also, the superscripts on the name of the models are not defined in this figure, or anywhere near it - they have to be found in the caption on Figure 2 (formerly Figure 3)
> > >
> > > Regarding the Optic Flow figure, I appreciate the author’s effort in adding more examples of different frames in an effort to generalize beyond a single example. The explanation given here is helpful but the manuscript still lacks a sufficient description of it. I believe a reader would still be unable to understand the choice of these frame pairs, and my questions “why this is useful to visualize in terms of improving understanding of representational alignment to human fMRI data?” remain unanswered in the manuscript. Furthermore, there are no quantitative metrics for the vague phrases ‘very different loadings along the second principal component’, and ‘large differences in the overall amount of optic flow’. This experiment really needs both better justification and a quantitative grounding.
> > >
> > > In the future, it would be helpful to the reviewers to mark changes to the updated manuscript in a different color to highlight them.
> > >
> > > Given there are improvements but remaining issues, I raise my score to a 6.

---

> > > > ### Author Response · Authors · 2024-11-28
> > > > **Response to Reviewer Xyvz - round 2**
> > > >
> > > > We are thankful for the positive assessment of the improvements as well as for the additional constructive feedback. The suggestion of using tSNE and UMAP is helpful, we will try to see if it is possible to add it to the camera ready version of the article at this stage. Our main rationale was opting for PCA was the dependence of t-SNE on parameters such as perplexity, that increase experimenter degrees of freedom. Nonetheless, we agree that the advantages of t-SNE in terms of its ability to capture nonlinear structure are important, and including both PCA and t-SNE could offer a more comprehensive picture of the results.
> > > >
> > > > The motivation for the claim that the difference in loadings on PC2 indicates a fundamentally different representational geometry in the optic flow models is that the principal component analysis was performed using as points the RDMs of the models’ layers. As a consequence, in the visualized space, each point corresponds to a model’s layer, and the distance between points corresponds to the difference between the RDMs in those layers. Each dimension in the space can be thought of as a “basis” RDM, and the loading of a layer on that dimension can be thought of as the amount of that “basis” RDM that needs to be used to reconstruct the RDM in that layer. The fact that the optic flow models are distant from the other models along PC2, therefore, means that the RDMs in the optic flow models are different from the RDMs in the other models, and that to a degree they are different “in the same way”: in the sense that a larger amount of the “basis” RDM corresponding to PC2 needs to be used to reconstruct the RDMs in those layers. We agree that a more quantitative analysis of the size of PC2 loadings as well as the differences in optic flow would strengthen the manuscript, unfortunately we were not able to complete these analyses within the time available, we will work on them in the future. We appreciate the Reviewer’s helpful suggestions and thank them for their contribution to improving the paper.

---

### Official Review · Reviewer_mTch · 2024-11-08

**Soundness:** 2
**Presentation:** 2
**Contribution:** 2
**Rating:** 5
**Confidence:** 3

**Summary:**

This paper aims to study the effect of neural alignment between a wide set of computer vision models such as Masked Auto Encoders and other computer vision models including how they are trained and whether they include temporal information as well. Authors aim to show through a set of experiments that models that include temporal information are more representationally aligned with real human neural activity extracted from fMRI.

**Strengths:**

* Authors aim to tackle a highly relevant problem which is the effect of static vs dynamic computer vision models (trained on single frames vs video data with a temporal component) with real brain data.
* Another strength of the paper is the use of real brain imaging data to assess model correspondence when shown a set of video frames.

**Weaknesses:**

I think the analysis done in this paper do not correspond to the papers' main conclusion. For example, the conclusion at the end of the paper that states that computer vision dynamic models are more neurally aligned with brain data vs static models is not really supported in Figure 4 or Figure 5. In Figure 4, the RDM brain trajectories are of several models plotted compared against each other.

I would have excepted to see a line as well for the "visual cortex" all-together (human ground truth, and incrementally from V1,V2,V4, IT etc...) so that we can qualitatively make an assessment of human vs machine alignment. Further figure 5 also seems strange: What does the 1st and 2nd PC across frames have to do with saying that dynamic models are better than static ones.

It overall feels like the RDM analysis is done incorrectly. I am under the impression that the figures I would be ideally looking at is comparing human vs machine feature outputs or or recording for a collection of visual stimuli. Instead, models are being compared to each other given their activation *per brain region*. So it somehow feels like the analysis was being done within vs between systems.

The title seems too strong for the claim and methods that are being used. Models used are mainly MAE's and masked video distillation, though table 1 does show a larger family of models with some lack of details. Which supervised model or self-supervised and why not explicitly state such models like it is done for MAEs?

**Questions:**

* I'd like to get more clarity of Figure 4 and Figure 5. In Figure 4 it is not obvious where the trajectory begins and ends. Where is the first layer and where is the last? It seems like it is plotted 0 and 10, but it's very cluttered in Figure 4. Which model should be the ground truth and how can you plot human ground truth as well if there are a different number of layers of processing between humans and machines.

* Also what should I be looking at in Figures 1,2 and 3. How can "model victory" of one vs all be declared in ideal conditions so I can better understand what to look for in all these figures? What would the ideal output look like? Should a model get to the noise ceiling for all brain areas? I'm under the impression that this will also depend on the layer of the neural network that is being compared, to the point that each layer of an artificial neural network must be compared to each area of the biological neural network, as correlations will differ and vary based on depth.

**Details Of Ethics Concerns:**

No details of ethical concerns as far as I am concerned, authors mention from where they collect their brain-data (previously open-sourced data repository)

---

> ### Author Response · Authors · 2024-11-27
> **Response to Reviewer mTch**
>
> >>> Weaknesses:
> >>> I think the analysis done in this paper do not correspond to the papers' main conclusion. For example, the conclusion at the end of the paper that states that computer vision dynamic models are more neurally aligned with brain data vs static models is not really supported in Figure 4 or Figure 5. In Figure 4, the RDM brain trajectories are of several models plotted compared against each other.
> >>>I would have excepted to see a line as well for the "visual cortex" all-together (human ground truth, and incrementally from V1,V2,V4, IT etc...) so that we can qualitatively make an assessment of human vs machine alignment. Further figure 5 also seems strange: What does the 1st and 2nd PC across frames have to do with saying that dynamic models are better than static ones.
> >>> It overall feels like the RDM analysis is done incorrectly. I am under the impression that the figures I would be ideally looking at is comparing human vs machine feature outputs or or recording for a collection of visual stimuli. Instead, models are being compared to each other given their activation per brain region. So it somehow feels like the analysis was being done within vs between systems.
>
> We appreciate Reviewer mTch’s constructive feedback, and we realize that our writing in the first version of the manuscript lacked clarity. When Reviewer mTch notes “I am under the impression that the figures I would be ideally looking at is comparing human vs machine feature outputs or or recording for a collection of visual stimuli.”, they are indeed correct. The figures comparing human recordings to machine outputs are Figures 1, 2 and 3  (now consolidated into Figures 1 and 2).
>
> The manuscript is organized into two sets of analyses. The first set of analyses, corresponding to Figures 1, 2 and 3 (now 1 and 2), compares the computer vision models to brain data. The remaining analyses investigate more closely the differences between the models that explain unique variance in neural responses (s-sup dynamic and sup dynamic) and all other models. Former Figures 4 and 5 (now 3 and 4) indeed do not contain comparisons with brain data, because those comparisons have been completed in the analyses shown in the previous Figures.
>
> We have added a section to the Methods explaining more clearly the brain-model comparisons, titled “Comparison between models and neural responses”. The section includes the following explanation of the brain-model comparisons:
>
> “Models were compared to neural responses using Representational Dissimilarity Matrices (RDMs, \cite{kriegeskorte2008representational}). In this study, RDMs are matrices whose rows and columns correspond to timepoints in the movie, such that the element of the matrix at a given row and column is the dissimilarity between the representation of the video at the timepoints that correspond to that row and column. Neural RDMs and model RDMs were compared by computing their Pearson correlation.
>
> The match between neural RDMs and RDMs for an entire model were calculated by first computing RDMs for each layer of the model and then computing a linear combination of the layer RDMs that best matches the neural RDM. In order to prevent circularity in the analysis, the weights attributed to each layer in the linear combination were calculated using 7 of the 8 experimental runs, and were applied to the model RDMs in the left-out run to compute a “predicted” RDM. We then evaluated the correlation between the “predicted” RDM and the neural RDM in the corresponding run (Figure 1).”
>
> The goal of the analysis in Figure 2 (which is now Figure 2a after the revision) was to determine whether combining multiple models provides a better account of neural responses compared to using a single model. We added this explanation to the Methods section:
>
> “To evaluate the joint neural predictivity for each pair of models, we computed a linear combination of the layer RDMs from both models that best matches neural responses. The weights for each layer were calculated using 7 of the 8 experimental runs, and the correspondence between the models and neural responses was then evaluated on the left-out run. This analysis enabled us to test whether and to which extent using two models jointly yielded a better match to neural responses than using a single model.”
>
> And regarding Figure 3 (which is now Figure 2b after the revision):
>
> “Finally, we wanted to test more directly the unique variance in a neural RDM that was explained by a model above and beyond each other model. To compute this, we regressed out a control model RDM from a neural RDM, and predicted the residual neural RDM with a target model, obtaining the unique variance explained by the target model. Matrices in Figure 2b show these difference values, with the target models as the columns and the control models as the rows.”

---

> > ### Author Response · Authors · 2024-11-27
> > **Response to Reviewer mTch - part 2**
> >
> > Because the human brain is organized into different regions with different functional roles, we compared the models to each different brain region separately, because we anticipated that some brain regions might show greater correspondence with some models, and other brain regions might show greater correspondence with other models. However, the results were largely consistent across all different regions, revealing that some models are more aligned with brain data overall. In particular, models that represent optic flow information (s-sup dynamic and sup dynamic) explain unique variance in neural responses that is not captured by the other models. This is shown in Figure 3 (now Figure 2b), specifically, the larger values in columns 3 and 4 of the matrices in Figure 3 indicate that the s-sup dynamic and the sup-dynamic models account for unique variance in the neural responses after controlling for the other models. As recommended by Reviewer mTch, Figures 1 to 3 (now consolidated into Figures 1 and 2) are “comparing human vs machine feature outputs or recordings for a collection of visual stimuli”. We have added the following passage to the article to explain more clearly what can be concluded from the brain-model comparisons:
> >
> > “As a key takeaway, the results show that the CNN models using optic flow (namely, s-sup dynamic and sup dynamic) explain unique neural variance that is not captured by MAE models (Figure 2b, columns 3-5 and rows 7-12 of the matrices). Importantly, they also explain unique variance that is not captured by other CNNs – even when they are trained with the same dataset (HAA, Figure 2b, columns 4-5, row 2 of the matrices). The results therefore indicate that the difference between CNN and Transformer architectures alone is not sufficient to account for the unique variance in neural responses explained by the models using optic flow.”
> >
> > The second set of analyses focused on comparing the models between each other, with the goal of identifying differences between the models that explained unique variance in neural responses (s-sup dynamic and sup dynamic) and the other models. Thus, Figure 4 and 5 (now 3 and 4) do not compare the models to brain activity: the comparison between models and brain activity is shown in Figures 1, 2 and 3  (now consolidated into Figures 1 and 2). In the original version of the manuscript, the rationale and interpretation of Figures 4 and 5 (now 3 and 4) was not adequately explained. We have addressed this limitation including the following passage in the Results section to explain the interpretation of Figure 4 (now renamed as Figure 3)
> >
> > “the results of principal component analysis (Figure 3) reveal that layers in the models using optic flow representations encode representations with a fundamentally different representational geometry compared to the other models. This is evidenced by the higher loadings of the optic flow models on the second principal component. By contrast, layers in the MAE models as well as in the CNNs that do not use optic flow information have lower loadings on the second principal component.”

---

> > > ### Author Response · Authors · 2024-11-27
> > > **Response to Reviewer mTch - part 3**
> > >
> > > Regarding Figure 5 (now renamed as Figure 4), we wanted to obtain a more interpretable understanding of the stimulus features driving the differences between the models that use optic flow and the rest of the models. We noted that the models that use optic flow mostly differ from the rest of the model along the second principal component (PC2). Principal components have loadings on each element of the representational dissimilarity matrix. Since the representational dissimilarity matrix contains the dissimilarity between different timepoints in the video, we could identify which pairs of timepoints in the video have high loadings for PC2. We expanded the figure to include 8 example pairs of timepoints with high loadings for PC2, showing the frames as well as optic flow maps at those timepoints. Consistently across the examples, one timepoint in the pair is characterized by more optic flow than the other, further supporting the hypothesis that the differences observed between the models are driven by how they represent optic flow information. We added the following caption to explain the findings:
> > > “Visualization of pairs of frames with very different loadings along the second principal component in the space of the models’ representational dissimilarity matrices. Each row illustrates the frames’ appearance and their optic flow. Images with different loadings along the second principal component typically show large differences in the overall amount of optic flow.”
> > > We interpret the results as indicating that differences in optic flow representation distinguish between the models that account for unique neural variance (s-sup dynamic and sup dynamic) and the rest of the models. We edited the manuscript to state this explicit in the Conclusion:
> > >
> > > “In conclusion, three models in particular explained unique variance in the neural responses that was not captured by other models: s-sup dynamic trained with Kinetics, s-sup dynamic trained with HAA and sup dynamic trained with HAA. These are the three models that represent optic flow information. The results converge to indicate that the use of optic flow information is a key factor that differentiates these models from the other models tested.”
> > > Note that this passage references three models because we have added a self-supervised dynamic CNN trained with the Kinetics dataset, in order to expand the range of models tested. Importantly, this new analysis enables a direct comparison between the self-supervised dynamic CNN trained with Kinetics and the Video MAEs that were also trained with Kinetics, showing that the self-supervised CNN captures unique variance in neural responses that is not explained by the Video MAEs, and ultimately that the difference between Video MAEs and self-supervised CNNs is not just driven by differences in the visual diet.
> > >
> > > >>> The title seems too strong for the claim and methods that are being used. Models used are mainly MAE's and masked video distillation, though table 1 does show a larger family of models with some lack of details. Which supervised model or self-supervised and why not explicitly state such models like it is done for MAEs?
> > >
> > > We thank the Reviewer for this helpful suggestion to improve the clarity of the manuscript. The Methods section in the revised manuscript lists each model with references to the articles in which they were introduced. Specifically, we include all branches of the Hidden Two Stream network model (spatial stream, TinyMotionNet and flow classifier), Image MAEs, Video MAEs, and Masked Video Distillation. The supervised static models are convolutional ResNets using video frames as the input, the self-supervised dynamic models are based on the TinyMotionNet architecture in Hidden Two Stream networks, and the supervised dynamic models are convolutional ResNets using optic flow as the input. In addition, we have included further analyses with a self-supervised dynamic CNN (TinyMotionNet architecture) trained on the Kinetics dataset (mentioned earlier in our response), as well as a new self-supervised vision Transformer that uses a different learning objective from MAEs (Dyno v2).

---

> > > > ### Author Response · Authors · 2024-11-27
> > > > **Response to Reviewer mTch - part 4**
> > > >
> > > > >>> Questions:
> > > >
> > > > >>> I'd like to get more clarity of Figure 4 and Figure 5. In Figure 4 it is not obvious where the trajectory begins and ends. Where is the first layer and where is the last? It seems like it is plotted 0 and 10, but it's very cluttered in Figure 4. Which model should be the ground truth and how can you plot human ground truth as well if there are a different number of layers of processing between humans and machines.
> > > >
> > > > Due to the large number of models and layers, it is difficult to eliminate clutter in Figure 4 (now renamed as Figure 3). However, importantly, the key information in the Figure concerns the layers and models that stand out from the clutter: the red, green and purple lines. These correspond to layers in the models that use optic flow, and are separated from the rest of the models that cluster together. These are also the models that predict unique variance in neural responses compared to the other models, as shown in Figure 3 (now Figure 2b).
> > > > The key takeaway from the Figure is that the models that explain unique neural variance (as shown by columns 3 to 5 of the matrices in Figure 2b) also differ from the other models along the second principal component (PC2). Figure 5 (now renamed as Figure 4) then aims to identify the stimulus properties that are associated with PC2, showing that pairs of timepoints that differ in terms of their PC2 loadings also vary greatly in terms of the optic flow at those timepoints, further supporting the conclusion that the models that explain unique neural variance differ from the other models in terms of their optic flow representations.
> > > >
> > > > >>> Also what should I be looking at in Figures 1,2 and 3. How can "model victory" of one vs all be declared in ideal conditions so I can better understand what to look for in all these figures? What would the ideal output look like? Should a model get to the noise ceiling for all brain areas? I'm under the impression that this will also depend on the layer of the neural network that is being compared, to the point that each layer of an artificial neural network must be compared to each area of the biological neural network, as correlations will differ and vary based on depth.
> > > > One possible ideal result is to identify one model that achieves noise ceiling for all brain areas. However, alternatively, it is possible that different models will be needed to account for the responses in different areas. In either case, an ideal situation would be to obtain for each brain area at least one model that achieves noise ceiling for that area. Whether it is the same model across areas or not.
> > > >
> > > > We concur with Reviewer mTch that correlations between brain responses and neural network layers might vary from layer to layer: further analyses could investigate individual layers separately. In this work we focused on analyzing each model as a whole, calculating the cumulative contribution of all the different layers in the model. A Figure showing a measure of “model victory” for one model vs other models is former Figure 3 (now 2b): higher values in the matrix indicate that the model in the column accounts for unique neural variance after controlling for the model in the row. In other words, this Figure indicates model victory for models in column 3 to 5 (that are associated with higher values, illustrated by the vertical yellow band at those two columns). These are the models that include representations of optic flow (s-sup dynamic and sup dynamic).
> > > >
> > > > We would like to thank Reviewer mTch for their questions and suggestions, we believe that the edits have improved the manuscript compared to the original version. We hope the Reviewer will consider updating their score in light of the revisions.

---

> > > > > ### Author Response · Authors · 2024-12-02
> > > > > **Follow up**
> > > > >
> > > > > As today is the last day that reviewers may post a message to the authors, we welcome any acknowledgment of our revisions and any additional feedback. Thank you.

---

### Official Review · Reviewer_Wd8d · 2024-11-09

**Soundness:** 3
**Presentation:** 3
**Contribution:** 2
**Rating:** 6
**Confidence:** 4

**Summary:**

In this paper author compare visual representations in masked autoencoders (MAEs) and video MAEs with human fMRI responses.

In the first experiments, they find that representation of dynamic models (VideoMAE + Video CNNs) that take into account temporal information is more correlated with human fMRI responses as compared to models trained with static information.

In the follow-up experiments using unique variance analysis, the authors find that convolutional models trained using optical flow explained unique variance in fMRI responses which was not explained by any other static/dynamic model suggesting the importance of optical flow loss in high correspondence with the fMRI responses.

Overall this paper finds that high performing MAEs on vision task show low correspondence with human fMRI responses as compared to standard CNNs.

**Strengths:**

1. This paper investigates an important question whether masked image/video modeling which leads to high performing vision models leads to representations that are better aligned with human brain responses.
2. The authors perform deep dive into which models are explaining variance in fMRI responses uniquely to determine the individual contribution of the models investigated. This is crucial as representation of the models is highly correlated.
3. The paper is easy to follow and clearly written. The figures are clearly explained in captions.
4. Comprehensive architecture choices: e.g. MAE after/before finetuning on Imagnet and VideoMAE after/before finetuning on Kinetics show that even after finetuning on Imagenet the MAE shows lower correlation than resnet trained on Imagenet

**Weaknesses:**

1. Results are reported on a single fMRI dataset. With large scale public fMRI datasets e.g. NSD, Algonauts videos available, the authors could have reported results on multiple dataset and showed generalizability of their results
2. I am not sure how different are individual conditions in the RDMs. If the conditions are part of the same movie then there could be high correlation between the individual conditions
3. Lack of interpretations of the results. While the authors shows model X show less correlation than model Y. An interpretation of why this could be happening is missing in the text and its relevance to brain regions functions

**Questions:**

1. Why video CNNs were trained on a different dataset as compared to video MAEs
2. How do authors infer that different types of dynamic information are represented in
OFA (face-selective), EBA (body-selective), and TOS (scene-selective). Line 367
3. I was not able to understand what is happening in Figure 4 and Figure 5 (Section 3.2). What was the motivation to do this analysis? How was this analysis performed and what was the result and interpretation?
4. What is the dimension of RDM? How did you divide movie segments ?

---

> ### Author Response · Authors · 2024-11-27
> **Response to Reviewer Wd8d**
>
> We thank Reviewer Wd8d for the positive evaluation of the manuscript, and report our responses below.
>
> >>> Weaknesses:
> >>> Results are reported on a single fMRI dataset. With large scale public fMRI datasets e.g. NSD, Algonauts videos available, the authors could have reported results on multiple dataset and showed generalizability of their results
>
> We agree with Reviewer Wd8d about the importance of the generalizability and robustness of the results. Due to the time constraints for resubmission we were not able to redo the analyses on an entirely new dataset. One aspect we would like to note, however, is that the prediction of neural RDMs was computed by estimating weights with 7 of the 8 experimental runs, and by evaluating the brain-model correlation using data from the left-out run. While this does not fully address the Reviewer’s comment, we believe that due to the differences between different sections of the movie it helps to support the generalizability of the results.
>
> >>> I am not sure how different are individual conditions in the RDMs. If the conditions are part of the same movie then there could be high correlation between the individual conditions
>
> In the RDMs, each row/column corresponds to a different timepoint in the movie. Some pairs of timepoints are very similar, while others are very different, depending on the similarity between the scenes depicted at those different timepoints. The movie includes a wide variety of different scenes, that include indoor and outdoor scenes as well as scenes with a low and high amount of movement, leading to a broad range of correlation values.
>
> >>> Lack of interpretations of the results. While the authors shows model X show less correlation than model Y. An interpretation of why this could be happening is missing in the text and its relevance to brain regions functions
>
> We concur with Reviewer Wd8d that the original version of the manuscript did not adequately discuss the interpretation of the results. To address this limitation, we have made the following additions to the Results and Discussion sections of the article (Figures 2,3 were consolidated to Figure 2, and Figures 4,5 renamed to Figures 3,4):
>
> “As a key takeaway, the results show that the CNN models using optic flow (namely, s-sup dynamic and sup dynamic) explain unique neural variance that is not captured by MAE models (Figure 2b, columns 3-5 and rows 7-12 of the matrices). Importantly, they also explain unique variance that is not captured by other CNNs – even when they are trained with the same dataset (HAA, Figure 2b, columns 4-5, row 2 of the matrices). The results therefore indicate that the difference between CNN and Transformer architectures alone is not sufficient to account for the unique variance in neural responses explained by the models using optic flow.”
>
> “The additional unique variance explained by the optic flow models (s-sup dynamic and sup dynamic) varied across regions, being strongest in EBA and TOS and weakest in FFA. The effect was observed widely, in regions previously associated with the processing of dynamic information (such as STS), but also in ventral temporal regions that have not been typically associated with the representation of dynamics (such as PPA). This observation is consistent with recent work suggesting that dynamic information is represented in a broader range of brain regions than previously thought (\cite{robert2023disentangling}).”
>
> “In other words, the results of principal component analysis (Figure 3) reveal that layers in the models using optic flow representations encode representations with a fundamentally different representational geometry compared to the other models. This is evidenced by the higher loadings of the optic flow models on the second principal component. By contrast, layers in the MAE models as well as in the CNNs that do not use optic flow information have lower loadings on the second principal component.”
>
> “It is important to note however that the representation of optic flow is likely not the only contributor to the worse alignment of the MAEs. In fact, even when trained with an action dataset (HAA), the static CNN showed better alignment with neural responses compared to the Image MAEs, and accounted for unique neural variance that was not captured by the MAEs.”
> “In conclusion, three models in particular explained unique variance in the neural responses that was not captured by other models: s-sup dynamic trained with Kinetics, s-sup dynamic trained with HAA and sup dynamic trained with HAA. These are the three models that represent optic flow information. The results converge to indicate that the use of optic flow information is a key factor that differentiates these models from the other models tested.”

---

> ### Author Response · Authors · 2024-11-27
> **Response to Reviewer Wd8d - part 2**
>
> We also clarified the key take home message in the Conclusion:
>
> “Convolutional models based on optic flow explained unique variance in neural responses that was not accounted for by any other model, not even video MAEs. Analysis of the representational geometry in the different layers of the models revealed that the second principal component in the space of representational dissimilarity matrices (RDMs) distinguished between convolutional models based on optic flow on one hand (which scored highly on the component) and all the other models on the other hand, suggesting a critical role of optic flow representations in human vision. We probed this conclusion further by examining the loadings of this component, and identifying pairs of scenes in the movie that were differentiated by the models based on optic flow but not by the other models. These included scenes with similar entities and backgrounds, that differed in the presence or absence of overall background flow (e.g. due to movement of the camera), further supporting the conclusion that video MAEs do not encode a set of dynamic features that are instead computed by both optic flow models and by human vision.”
>
> And
>
> “The difference in alignment with neural responses between MAEs and CNNs  is likely also driven in part by additional factors above and beyond optic flow. In particular, the comparison between ImageMAEs and the static net trained with ImageNet indicates that differences in architecture and task also play an important role for the differences in alignment with neural responses.”
>
> And
>
> “In conclusion, the results converge to indicate that the lack of optic flow representations and the use of self-supervised Vision Transformer architectures are jointly responsible to account for decreased alignment between models and neural representations.”
>
>
> >>> Questions:
> >>> Why video CNNs were trained on a different dataset as compared to video MAEs
>
> To address this limitation, we have additionally trained a video CNN using Kinetics, the same dataset used to train Video MAEs. The video CNN trained with Kinetics yielded a pattern of results analogous to the video CNN trained on HAA, indicating that these results are not due to differences in the training dataset.
>
> >>> How do authors infer that different types of dynamic information are represented in OFA (face-selective), EBA (body-selective), and TOS (scene-selective). Line 367
>
> We meant to say that dynamic information contributes more to accounting for EBA and TOS responses, this is based on the higher values in Figure 3, columns 3-4 for those regions. We have amended the text accordingly.

---

> > ### Author Response · Authors · 2024-11-27
> > **Response to Reviewer Wd8d - part 3**
> >
> > >>> I was not able to understand what is happening in Figure 4 and Figure 5 (Section 3.2). What was the motivation to do this analysis? How was this analysis performed and what was the result and interpretation?
> >
> > We apologize for the lack of clarity regarding these figures in the original version of the manuscript. We have now added a more detailed explanation of the rationale behind these analyses and of the interpretation of the results.
> >
> > Figures 4 and 5 (now 3 and 4) were aimed at investigating in more depth the differences between models that explain unique neural variance and other models. To investigate this in more detail, we treated the representational dissimilarity matrix of each model’s layer as a high-dimensional vector. Therefore, analyzing all models and layers, we obtained a set of vectors, and performed principal component analysis (PCA) to identify dimensions that account for most variation in the representations encoded in different layers of different models (Figure 3). In line with the observations noted in the previous paragraph, we found that the models that use optic flow learn very different representational structure compared to both MAE models and other CNNs (see the green, red and purple lines). This observation converges with the argument outlined in the previous paragraph to indicate that the differences are not due to an overall distinction between CNN and Transformer architectures. Instead, among CNNs, those that use optic flow learn different representations compared to all other models, including video MAEs.
> >
> > At this point, we wanted to obtain a more interpretable understanding of the stimulus features driving the differences between the models that use optic flow and the rest of the models. We noted that the models that use optic flow mostly differ from the rest of the model along the second principal component (PC2). Principal components have loadings on each element of the representational dissimilarity matrix. Since the representational dissimilarity matrix contains the dissimilarity between different timepoints in the video, we could identify which pairs of timepoints in the video have high loadings for PC2. We visualized 8 example pairs of timepoints with high loadings for PC2 in Figure 4, showing the frames as well as optic flow maps at those timepoints. Consistently across the examples, one timepoint in the pair is characterized by more optic flow than the other, further supporting the hypothesis that the differences observed between the models are driven by how they represent optic flow information.
> >
> > We have modified the text to explain more clearly our interpretation of the results and the logic that drives it. Specifically, we have added the following passage to the Results section to highlight that optic flow CNNs account for unique variance in the neural responses not only compared to MAEs but also compared to static CNNs:
> >
> > “As a key takeaway, the results show that the CNN models using optic flow (namely, s-sup dynamic and sup dynamic) explain unique neural variance that is not captured by MAE models (Figure 2b, columns 3-5 and rows 7-12 of the matrices). Importantly, they also explain unique variance that is not captured by other CNNs – even when they are trained with the same dataset (HAA, Figure 2b, columns 4-5, row 2 of the matrices). The results therefore indicate that the difference between CNN and Transformer architectures alone is not sufficient to account for the unique variance in neural responses explained by the models using optic flow.”
> >
> > And the following passage to highlight that optic flow models deviate from other CNNs as well as MAEs in the principal component analysis results:
> >
> > “In other words, the results of principal component analysis (Figure 3) reveal that layers in the models using optic flow representations encode representations with a fundamentally different representational geometry compared to the other models. This is evidenced by the higher loadings of the optic flow models on the second principal component. By contrast, layers in the MAE models as well as in the CNNs that do not use optic flow information have lower loadings on the second principal component.”
> >
> > Furthermore, we have expanded Figure 4 to illustrate a broader set of examples of timepoints that differ along the dimension that drives the separation between optic flow models and the other models, and added the following caption to explain the findings:
> >
> > “Visualization of pairs of frames with very different loadings along the second principal component in the space of the models’ representational dissimilarity matrices. Each row illustrates the frames’ appearance and their optic flow. Images with different loadings along the second principal component typically show large differences in the overall amount of optic flow.”

---

> > > ### Author Response · Authors · 2024-11-27
> > > **Response to Reviewer Wd8d - part 4**
> > >
> > > >>> What is the dimension of RDM? How did you divide movie segments ?
> > > Movie segments were divided to obtain sections of approximately equal length. The dimension of the RDMs obtained for the eight segments were 451, 441, 438, 488, 462, 439, 542, 338. We have now added this information to the manuscript.

---

> > > > ### Author Response · Authors · 2024-12-02
> > > > **Follow up**
> > > >
> > > > As today is the last day that reviewers may post a message to the authors, we welcome any acknowledgment of our revisions and any additional feedback. Thank you.

---

### Official Review · Reviewer_kFNo · 2024-11-11

**Soundness:** 3
**Presentation:** 3
**Contribution:** 2
**Rating:** 6
**Confidence:** 4

**Summary:**

This work provides empirical and comprehensive studies of the alignment between artificial neural networks' representations and the brain's representations. The motivation comes from the generalist performance of MAE; therefore, MAE is expected to have similar representations to the brain. However, MAE has poorer alignment with the brain even when compared with supervised CNNs. Supervised finetuning does improve the alignment between MAE and the brain but it is not enough. Moreover, the authors also suggest that temporal information could make artificial neural networks more aligned with the human brain.

**Strengths:**

The alignment between neural networks and the brain is the topic that is very worth studying and the authors show that even the strong vision transformer models, that can be finetuned to many vision and control tasks, are not really aligned with the brain. The work is novel and could give insights to theoretical and computational neuroscience.

**Weaknesses:**

1. The conclusion of the paper does not lead to any scientific theory or a glimpse of it. Specifically, it is unclear whether the misalignment between MAE and human brains is due to Transformer Architecture or the Masked Autoencoder pre-training itself. The study has to be more concrete to conclude the theory. To decouple the confounders, I suggest the authors do more experiments on Masked Autoencoder with CNN and see if the low neural alignment still exists. The authors may also play around with different “noises” as Masked Autoencoder is a special case of Denoising Autoencoder. If Masked CNN is much more aligned with the brain, it is probably because of Transformer architecture that causes misalignment. Last but not least, the authors may use other kinds of pre-trained vision transformers that work equally well compared to MAE pre-training, such as MoCo v3, Dino v2. If Transformer is really misaligned with the brain, I would expect a low alignment regardless of the pre-training method. If it is because of the Masked Autoencoder, I would expect high alignment.


2. There is no control over visual diets. Comparing different architectures and training paradigms should be done on the same visual diets. For example, comparing CNN trained on an action recognition dataset with MAE trained on ImageNet cannot tell anything. I suggest separating experiments based on visual diets.


3. The authors only consider high-level areas of the brain. There is no comparison of the artificial neural networks with lower visual areas like V1, V2 and V4 which are important for visual object recognition.

**Questions:**

1. Regarding Fig 1, what do authors mean by "Predicted RDM"? RDM is calculated by responses of the networks across categories of stimuli. How could you predict RDM? Moreover, the definition of RDM should be written, at least in appendices. General audience may not understand it.

2. How are matrices in Fig 2 and Fig 3 computed? I cannot fully understand the intuition behind these matrices. If the authors use the methods existing in the previous works, they should be mentioned so that the readers can understand the full details.


3. The paper says that “Unlike VMAE and MAE, the MVD model does not learn pixel-level features.” Is this true? The MVD model takes ‘pixels' as input, so it should learn pixel-level features. The authors’ meaning may be that MVD does not use 'pixel errors' as learning signals, which is a distinct approach compared to models like VMAE and MAE. Please correct me if I am wrong.

4. The first set of analyses 3 quantify the correspondence between different models and category-selective brain regions.
What is analyses 3, is it a typo?

---

> ### Author Response · Authors · 2024-11-27
> **Response to Reviewer kFNo**
>
> >>> Weaknesses:
> >>> The conclusion of the paper does not lead to any scientific theory or a glimpse of it. Specifically, it is unclear whether the misalignment between MAE and human brains is due to Transformer Architecture or the Masked Autoencoder pre-training itself. The study has to be more concrete to conclude the theory. To decouple the confounders, I suggest the authors do more experiments on Masked Autoencoder with CNN and see if the low neural alignment still exists. The authors may also play around with different “noises” as Masked Autoencoder is a special case of Denoising Autoencoder. If Masked CNN is much more aligned with the brain, it is probably because of Transformer architecture that causes misalignment. Last but not least, the authors may use other kinds of pre-trained vision transformers that work equally well compared to MAE pre-training, such as MoCo v3, Dino v2. If Transformer is really misaligned with the brain, I would expect a low alignment regardless of the pre-training method. If it is because of the Masked Autoencoder, I would expect high alignment.
>
> We thank Reviewer kFNo for this feedback, we agree that we could have been clearer about the scientific conclusions that can be obtained from the study. The scientific hypothesis we reach from this study is that the misalignment between MAE models and human brains might be due to how the human brain encodes information about optic flow. We discuss the details of our logic below. We have updated the Figures with additional analyses, the response refers to the updated Figures. Specifically, we have added results for early visual regions (V1-V4) in Figure 1 as Figure 1b, and we have combined the former Figures 2 and 3 into Figure 2 (as 2a and 2b).
>
> First, we observe that CNN models trained to learn optic flow (s-sup dynamic) as well as CNN models that use optic flow as input (sup dynamic) account for unique variance in neural responses that is not captured by the MAE models (see matrices in Figure 2b, columns 3 to 5 from the left, rows 6 to 12 from top). Importantly, CNN models that use optic flow information also account for unique variance in neural responses that is not captured by other CNN models (see matrices in Figure 2b, columns 3 to 5 from the left, rows 1 and 2 from top). This demonstrates that the CNN architecture on its own is not sufficient to account for the unique variance in neural responses, and that the improved correspondence with neural responses is restricted to the models that use optic flow (s-sup dynamic and sup dynamic).
>
> To investigate this phenomenon in more detail, we treated the representational dissimilarity matrix of each model’s layer as a high-dimensional vector. Therefore, analyzing all models and layers, we obtained a set of vectors, and performed principal component analysis (PCA) to identify dimensions that account for most variation in the representations encoded in different layers of different models (Figure 3). In line with the observations noted in the previous paragraph, we found that the models that use optic flow learn very different representational structure compared to both MAE models and other CNNs (see the green, red and purple lines). This observation converges with the argument outlined in the previous paragraph to indicate that the differences are not due to an overall distinction between CNN and Transformer architectures. Instead, among CNNs, those that use optic flow learn different representations compared to all other models, including video MAEs.
>
> At this point, we wanted to obtain a more interpretable understanding of the stimulus features driving the differences between the models that use optic flow and the rest of the models. We noted that the models that use optic flow mostly differ from the rest of the model along the second principal component (PC2). Principal components have loadings on each element of the representational dissimilarity matrix. Since the representational dissimilarity matrix contains the dissimilarity between different timepoints in the video, we could identify which pairs of timepoints in the video have high loadings for PC2. We visualized 8 example pairs of timepoints with high loadings for PC2 in Figure 3 (formerly Figure 4), showing the frames as well as optic flow maps at those timepoints. Typically, one timepoint in the pair is characterized by more optic flow than the other, further supporting the hypothesis that the differences observed between the models are driven by how they represent optic flow information.

---

> > ### Author Response · Authors · 2024-11-27
> > **Response to Reviewer kFNo - part 2**
> >
> > We have modified the text to explain more clearly our interpretation of the results and the logic that drives it. Specifically, we have added the following passage to the Results section to highlight that optic flow CNNs account for unique variance in the neural responses not only compared to MAEs but also compared to static CNNs:
> >
> > “As a key takeaway, the results show that the CNN models using optic flow (namely, s-sup dynamic and sup dynamic) explain unique neural variance that is not captured by MAE models (Figure 2b, columns 3-5 and rows 7-12 of the matrices). Importantly, they also explain unique variance that is not captured by other CNNs – even when they are trained with the same dataset (HAA, Figure 2b, columns 4-5, row 2 of the matrices). The results therefore indicate that the difference between CNN and Transformer architectures alone is not sufficient to account for the unique variance in neural responses explained by the models using optic flow.”
> >
> > And the following passage to highlight that optic flow models deviate from other CNNs as well as MAEs in the principal component analysis results:
> >
> > “In other words, the results of principal component analysis (Figure 3) reveal that layers in the models using optic flow representations encode representations with a fundamentally different representational geometry compared to the other models. This is evidenced by the higher loadings of the optic flow models on the second principal component. By contrast, layers in the MAE models as well as in the CNNs that do not use optic flow information have lower loadings on the second principal component.”
> >
> > Furthermore, we have expanded Figure 4 (formerly Figure 5) to illustrate a broader set of examples of timepoints that differ along the dimension that drives the separation between optic flow models and the other models, and added the following caption to explain the findings:
> >
> > “Visualization of pairs of frames with very different loadings along the second principal component in the space of the models’ representational dissimilarity matrices. Each row illustrates the frames’ appearance and their optic flow. Images with different loadings along the second principal component typically show large differences in the overall amount of optic flow.”
> >
> > Finally, we have clarified our interpretation of the overall pattern of results, stating more clearly our scientific hypothesis in the Conclusion section:
> >
> > “In conclusion, three models in particular explained unique variance in the neural responses that was not captured by other models: s-sup dynamic trained with Kinetics, s-sup dynamic trained with HAA and sup dynamic trained with HAA. These are the three models that represent optic flow information. The results converge to indicate that the use of optic flow information is a key factor that differentiates these models from the other models tested.”
> >
> >
> > As suggested by the Reviewer, we have additionally included Dino v2 in the analyses. As hypothesized by the Reviewer, Dino v2 performed similarly to other Image MAEs in terms of its correspondence with neural responses, suggesting that other Transformer architectures can also show worse alignment with neural responses. We have added the following passage to the results:
> >
> > “To further expand our investigation into the correspondence between neural representations and representations in vision Transformers, we additionally compared neural RDMs to the RDMs obtained with Dino v2, a self-supervised vision Transformer trained with a different self-supervised objective. The results revealed that worse alignment with neural responses was not restricted to Image MAEs, but extended to the Dino v2 model as well. This suggests that multiple types of self-supervised vision Transformers do not provide high correspondence with neural responses. More research will be needed to determine whether this result is due to the Transformer architecture itself.”

---

> > > ### Author Response · Authors · 2024-11-27
> > > **Response to Reviewer kFNo - part 3**
> > >
> > > There is no control over visual diets. Comparing different architectures and training paradigms should be done on the same visual diets. For example, comparing CNN trained on an action recognition dataset with MAE trained on ImageNet cannot tell anything. I suggest separating experiments based on visual diets.
> > >
> > > We agree with Reviewer kFNo that differences between the visual diet of some models is a limitation of this study. This limitation is partly due to the high cost of retraining the models on new datasets. We have attempted to mitigate this concern by training an additional model in the time available – an optic flow model (s-sup dynamic) trained with Kinetics (because the Video Masked Autoencoders were trained with Kinetics as well). The new results show that even when trained with Kinetics, the s-sup dynamic model accounts for unique neural variance compared to the Kinetics-trained Video MAEs (Figure 2b matrices, column 3, rows 10-12). This indicates that the difference in performance between s-sup dynamic and Video MAEs is not due to differences in the visual diet. We have added the following passage to the results section:
> > >
> > > “Importantly, the self-supervised dynamic model accounts for unique neural variance compared to the Video MAEs even when trained on the same dataset: Kinetics (Figure 2b matrices, column 3, rows 10-12). This indicates that the difference in performance between s-sup dynamic and Video MAEs cannot be fully attributed to differences in the visual diet.”
> > >
> > > The authors only consider high-level areas of the brain. There is no comparison of the artificial neural networks with lower visual areas like V1, V2 and V4 which are important for visual object recognition.
> > >
> > > We appreciate the suggestion to include earlier visual regions in our analysis. We have added the analyses for these regions to Figure 1. Specifically, we have analyzed the correspondence of the models with the following regions: V1 ventral, V2 ventral, V3 ventral, V1 dorsal, V2 dorsal, V3 dorsal, V4. The pattern of results is largely consistent with the findings observed in category-selective regions.
> > >
> > > >>> Questions:
> > > >>> Regarding Fig 1, what do authors mean by "Predicted RDM"? RDM is calculated by responses of the networks across categories of stimuli. How could you predict RDM? Moreover, the definition of RDM should be written, at least in appendices. General audience may not understand it.
> > >
> > > We appreciate these suggestions, we have included in the Methods section a subsection titled
> > > “Comparison between models and neural responses”
> > >
> > > The section includes the following explanation of RDMs:
> > >
> > > “Models were compared to neural responses using Representational Dissimilarity Matrices (RDMs, \cite{kriegeskorte2008representational}). In this study, RDMs are matrices whose rows and columns correspond to timepoints in the movie, such that the element of the matrix at a given row and column is the dissimilarity between the representation of the video at the timepoints that correspond to that row and column. Neural RDMs and model RDMs were compared by computing their Pearson correlation.”
> > >
> > > We also added the following passage, which addresses what we mean by saying that an RDM was “predicted”:
> > >
> > > “The match between neural RDMs and RDMs for an entire model were calculated by first computing RDMs for each layer of the model and then computing a linear combination of the layer RDMs that best matches the neural RDM. In order to prevent circularity in the analysis, the weights attributed to each layer in the linear combination were calculated using 7 of the 8 experimental runs, and were applied to the model RDMs in the left-out run to compute a “predicted” RDM. We then evaluated the correlation between the “predicted” RDM and the neural RDM in the corresponding run.”

---

> > > > ### Author Response · Authors · 2024-11-27
> > > > **Response to Reviewer kFNo - part 4**
> > > >
> > > > >>> 2.  How are matrices in Fig 2 and Fig 3 computed? I cannot fully understand the intuition behind these matrices. If the authors use the methods existing in the previous works, they should be mentioned so that the readers can understand the full details.
> > > >
> > > > The goal of the analysis in Figure 2 (which is now Figure 2a after the revision) was to determine whether combining multiple models provides a better account of neural responses compared to using a single model. We added this explanation to the Methods section:
> > > >
> > > > “To evaluate the joint neural predictivity for each pair of models, we computed a linear combination of the layer RDMs from both models that best matches neural responses. The weights for each layer were calculated using 7 of the 8 experimental runs, and the correspondence between the models and neural responses was then evaluated on the left-out run. This analysis enabled us to test whether and to which extent using two models jointly yielded a better match to neural responses than using a single model.”
> > > >
> > > > And regarding Figure 3 (which is now Figure 2b after the revision):
> > > >
> > > > “Finally, we wanted to test more directly the unique variance in a neural RDM that was explained by a model above and beyond each other model. To compute this, we regressed out a control model RDM from a neural RDM, and predicted the residual neural RDM with a target model, obtaining the unique variance explained by the target model. Matrices in Figure 2b show these difference values, with the target models as the columns and the control models as the rows.”
> > > >
> > > > >>> 3. The paper says that “Unlike VMAE and MAE, the MVD model does not learn pixel-level features.” Is this true? The MVD model takes ‘pixels' as input, so it should learn pixel-level features. The authors’ meaning may be that MVD does not use 'pixel errors' as learning signals, which is a distinct approach compared to models like VMAE and MAE. Please correct me if I am wrong.
> > > >
> > > > This is right, we have edited the paper to correct this point. The corrected version reads:
> > > > “Unlike VMAE and MAE, the MVD model does not use pixel-level errors as learning signals.”
> > > >
> > > > >>> 4. The first set of analyses 3 quantify the correspondence between different models and category-selective brain regions. What is analyses 3, is it a typo?
> > > >
> > > > Yes, this was a reference to a section of the article, we have edited to refer to the corresponding Figure to improve clarity.
> > > >
> > > > We thank reviewer kFNo for the extensive constructive feedback on our manuscript, which we believe helped us to improve it substantially. We hope the Reviewer will consider updating the manuscript’s score in light of the revisions.

---

> ### Comment · Reviewer_kFNo · 2024-11-27
>
> I appreciate the authors' responses. However, I do not think the manuscript is ready for publication. I would maintain my score to be 5 and improvements can be added for the next version. Moreover, I do not see the updated version of the manuscript, have you updated it or just prepared to update?
>
>
> 1. Response to: "The scientific hypothesis we reach from this study is that the misalignment between MAE models and human brains might be due to how the human brain encodes information about optic flow."
>
> I am afraid that this hypothesis is already falsified by some results in the main text. Referring to Fig.2 FFA, a sole supervised CNN can predict responses in this area and including other models do not improve the predictivity. Moving on to MAE, it cannot predict FFA responses well. This is not because of optical flow because CNN that predict the responses well with or without including optical flow representation from other CNN models. MAE has low predictivty for all regions regardless of the need for optical flow representation to better align with those regions.
>
>
>
>
> 2. This demonstrates that the CNN architecture on its own is not sufficient to account for the unique variance in neural responses, and that the improved correspondence with neural responses is restricted to the models that use optic flow (s-sup dynamic and sup dynamic).
>
> CNN or whatever architectures trained to predict optical flow would better explain responses in some brain areas that estimate the optical flow. I would not be surprised because CNN (sup dynamic) is trained to predict what some brain areas are doing. For some areas like FFA, there is no need for optical flows. Moreover, why is this related to the main hypothesis about MAE?
>
>
>
> 3. Response to explanation of Fig 2 and 3.
>
> In my opinion, Fig 2 and Fig 3 are highly dependent. Having only one of those figures would be enough to explain the "unique representations of the neural networks". I would like to know more about the cases where we need both Fig 2 and Fig 3 to explain the model. Correct me if I am wrong.
>
> For Fig 2, each element in the matrix comes from the "neural predictivity score" of combining two network models' responses in each row and column.
> Y = f(A,B) where f is the regression model and A and B are row and column model RDM and Y is the neural RDM.
>
> For Fig 3, each element comes from the "residual predictivity". To get the residual, the first model (row) predicts neural RDM and then residuals can be computed. Residual is then predicted by the second model (column) and the predictivity score can be computed based on how accurate the target model can predict the residual.
>
> Essentially, Fig2 and Fig3 matrices are from combining two models' responses (model RDM) to predict neural responses (neural RDM).

---

> > ### Author Response · Authors · 2024-11-27
> > **Response to Reviewer kFNo - round 2 - part 1**
> >
> > >>> I appreciate the authors' responses. However, I do not think the manuscript is ready for publication. I would maintain my score to be 5 and improvements can be added for the next version. Moreover, I do not see the updated version of the manuscript, have you updated it or just prepared to update?
> > Response to: "The scientific hypothesis we reach from this study is that the misalignment between MAE models and human brains might be due to how the human brain encodes information about optic flow."
> > >>> I am afraid that this hypothesis is already falsified by some results in the main text. Referring to Fig.2 FFA, a sole supervised CNN can predict responses in this area and including other models do not improve the predictivity. Moving on to MAE, it cannot predict FFA responses well. This is not because of optical flow because CNN that predict the responses well with or without including optical flow representation from other CNN models. MAE has low predictivty for all regions regardless of the need for optical flow representation to better align with those regions.
> > This demonstrates that the CNN architecture on its own is not sufficient to account for the unique variance in neural responses, and that the improved correspondence with neural responses is restricted to the models that use optic flow (s-sup dynamic and sup dynamic).
> > >>> CNN or whatever architectures trained to predict optical flow would better explain responses in some brain areas that estimate the optical flow. I would not be surprised because CNN (sup dynamic) is trained to predict what some brain areas are doing. For some areas like FFA, there is no need for optical flows. Moreover, why is this related to the main hypothesis about MAE?
> >
> > We have now uploaded the updated .pdf of the manuscript. We will comment on points 1 and 2 jointly as they are related. We agree with some parts of the Reviewer’s response.
> >
> > The part we disagree on is the hypothesis that CNNs trained to predict optical flow would better explain responses only in areas thought to estimate optical flow. Our interpretation of the results differs for the following reasons.
> >
> > 1) The models that use optic flow information explain unique additional variance not only in brain regions traditionally believed to represent optic flow information (such as regions in lateral temporal cortex), but also in ventral temporal regions that are not traditionally thought to encode optic flow information such as PPA and RSC (Figure 2b, row 1, columns 3-5). We also observe this effect in other ventral regions including VO1, VO2 (where VO stands for ventral occipital), PHC1 and PHC2 (where PHC stands for parahippocampal cortex) – we are currently working on a new figure including the results for these regions. This is consistent with recent work showing that ventral regions respond to kinematograms:
> >
> > Robert, Sophia, Leslie G. Ungerleider, and Maryam Vaziri-Pashkam. "Disentangling object category representations driven by dynamic and static visual input." Journal of Neuroscience 43.4 (2023): 621-634.
> >
> > Among 18 regions tested FFA shows a comparatively weaker unique contribution of the optic flow models compared to the static CNN trained with ImageNet, but the optic flow models do still explain some additional unique variance even in this region, as shown by the higher values in the FFA matrix in Figure 2b in the first row and columns 3-5 compared to the subsequent columns. While the size of the effect varies across regions, being strongest in TOS and EBA, the effect is not limited to regions traditionally believed to estimate optic flow information.
> >
> > 2) The static CNN the Reviewer refers to is trained with a different dataset compared to the optic flow models, ImageNet. A better comparison is offered by the same architecture trained with the same dataset as the optic flow models: sup static trained with HAA. When the same dataset is used to train both the static CNN and the optic flow CNNs, the optic flow CNNs explain more additional unique variance compared to the static CNN, even in FFA (Figure 2b, row 2, columns 3-5). Ideally, we would have a dataset that includes as broad a variety of objects as ImageNet, but that additionally includes dynamics, so that we could train the optic flow models and the MAEs on that dataset and compare them to the static CNN trained on ImageNet. However, critically, the hypothesis that optic flow plays an important role in driving differences in alignment with neural responses between MAEs and the s-sup and sup dynamic models is not falsified by the data: when the models are trained on comparable datasets (HAA, Kinetics), the optic flow models outperform MAEs as well as the static CNNs

---

> > > ### Author Response · Authors · 2024-11-27
> > > **Response to Reviewer kFNo - round 2 - part 2**
> > >
> > > We agree however with Reviewer kFNo on multiple points. First, the additional unique contribution of models using optic flow is not the same across all brain areas tested. In particular, the difference is strongest in EBA and TOS, and weakest in FFA, and this is an interesting pattern which is worth discussing. We added the following passage to the manuscript to make this point:
> > >
> > > “The additional unique variance explained by the optic flow models (s-sup dynamic and sup dynamic) varied across regions, being strongest in EBA and TOS and weakest in FFA. The effect was observed widely, in regions previously associated with the processing of dynamic information (such as STS), but also in ventral temporal regions that have not been typically associated with the representation of dynamics (such as PPA). This observation is consistent with recent work suggesting that dynamic information is represented in a broader range of brain regions than previously thought (\cite{robert2023disentangling}).”
> > >
> > > Second, we agree that additional factors beyond optic flow might contribute to the worse alignment of MAEs. For example, even when trained with a more comparable dataset (HAA), the sup static model explains unique additional variance compared to the Image MAEs as well as compared to the other transformer - Dyno v2 (Figure 2b, second column, rows 6-9). We have added the following sentence to the results to point this out:
> > >
> > > “It is important to note however that the representation of optic flow is likely not the only contributor to the worse alignment of the MAEs. In fact, even when trained with an action dataset (HAA), the static CNN showed better alignment with neural responses compared to the Image MAEs, and accounted for unique neural variance that was not captured by the MAEs.”
> > >
> > > We also highlighted this in the Conclusion:
> > >
> > > “The difference in alignment with neural responses between MAEs and CNNs  is likely also driven in part by additional factors above and beyond optic flow. In particular, the comparison between ImageMAEs and the static net trained with ImageNet indicates that differences in architecture and task also play an important role for the differences in alignment with neural responses.”
> > >
> > > And
> > >
> > > “In conclusion, the results converge to indicate that the lack of optic flow representations and the use of self-supervised Vision Transformer architectures are jointly responsible to account for decreased alignment between models and neural representations.”
> > >
> > > Response to explanation of Fig 2 and 3.
> > > >>> In my opinion, Fig 2 and Fig 3 are highly dependent. Having only one of those figures would be enough to explain the "unique representations of the neural networks". I would like to know more about the cases where we need both Fig 2 and Fig 3 to explain the model. Correct me if I am wrong.
> > > >>> For Fig 2, each element in the matrix comes from the "neural predictivity score" of combining two network models' responses in each row and column. Y = f(A,B) where f is the regression model and A and B are row and column model RDM and Y is the neural RDM.
> > > >>> For Fig 3, each element comes from the "residual predictivity". To get the residual, the first model (row) predicts neural RDM and then residuals can be computed. Residual is then predicted by the second model (column) and the predictivity score can be computed based on how accurate the target model can predict the residual.
> > > Essentially, Fig2 and Fig3 matrices are from combining two models' responses (model RDM) to predict neural responses (neural RDM).
> > >
> > > Yes, Figure 2 and 3 are closely related. This is why in the new version we opted to combine them into a single figure as Figures 2a and 2b.

---

> > > > ### Author Response · Authors · 2024-12-02
> > > > **Follow up**
> > > >
> > > > Considering the weaknesses raised by Reviewer kFNo in their original review:
> > > >
> > > > - for point 1. we have added Dino v2 as requested
> > > > - for point 2. we have added a s-sup dynamic model trained with Kinetics, matching the VMAEs trained with Kinetics in terms of the visual diet
> > > > - for point 3. we have added analyses for lower level areas as requested
> > > >
> > > > We have included point-by-point responses to the other comments. We believe that the revisions improved the article compared to the original version, and we welcome any additional feedback.

---

> > > > > ### Comment · Reviewer_kFNo · 2024-12-03
> > > > >
> > > > > I appreciate the authors's attempt to improve the manuscript presentation and the interpretation of the results.
> > > > > I would like to raise the score to 6 because this work provides useful information that could guide future research in human vision models in computational neuroscience. The result suggests that a self-supervised vision transformer is not a good model of human vision even after adding temporal information. It seems that not only MAE is a bad human vision model but also DinoV2 which is trained differently than MAE. The cause of the misalignment may be due to the inductive biases of vision transformers, not the training paradigm, to verify this we may need to evaluate transformers that are trained with supervised learning.
> > > > > I would suggest the authors to include the open problems that can be extended from this work in the discussion.

---

> > > > > > ### Author Response · Authors · 2024-12-03
> > > > > > **Response to Reviewer kFNo**
> > > > > >
> > > > > > We thank Reviewer kFNo for their feedback and we will work to include the open problems in the discussion as suggested.

---

### Meta-Review · Area_Chair_qwMA · 2024-12-22

**Metareview:**

This paper compares the representation layer of image and video MAEs to human fMRI response data for a variety of brain areas. The authors report several interesting findings, including that 1) MAEs diverge from neural representations in humans and convolutional neural networks, 2) video MAEs show closer correspondence which points to the importance of temporal information 3) convolutional networks based on optic flow show a closer correspondence to neural responses in humans than even video MAEs.

Strengths: Representational alignment of newer, high-performing models like MAEs with human cognition is both interesting and important. The worse alignment of MAEs is a surprising result that challenges the assumption that better performance would come with closer alignment. Comprehensive choice of architectures.
Weaknesses: Results are reported on a single fMRI dataset. There is little control regarding differences in  training data. The experiments do not suffice to pinpoint the cause of the decreased alignment (e.g. the role of the transformer architecture vs the training data / objective).

Overall this is a borderline paper. It presents surprising findings that would clearly be of interest to the community, but in its current state might not meet the bar for an ICLR paper. For now lean towards rejecting the paper but would definitely encourage the authors to polish their work a bit more and resubmit.

**Additional Comments On Reviewer Discussion:**

The authors made a commendable effort to incorporate the feedback of the reviewers, and the paper was clearly improved substantially during the discussion period. Sadly two of the reviewers did not respond to these changes which makes it difficult to judge if their concerns were sufficiently addressed.

---

### Decision · Program_Chairs · 2025-01-22

Reject